# Contactless Method for Measurement of Surface Roughness Based on a Chromatic Confocal Sensor

**Natalia Lishchenko \*, Garret E. O'Donnell and Mark Culleton**

Department of Mechanical, Manufacturing and Biomedical Engineering, Trinity College Dublin, College Green, Dublin 2, D02 PN40 Dublin, Ireland; garret.odonnell@tcd.ie (G.E.O.); culletom@tcd.ie (M.C.)
\* Correspondence: lishchen@tcd.ie

**Abstract:** The methodology for assigning and assessing the surface quality is used at various stages of the product life cycle: during the design and technological preparation of production, the production itself, and during the control (testing) of products. The development of advanced technologies requires in situ part control. A non-contact in situ surface roughness measuring system is proposed in this paper. The proposed system utilizes a chromatic confocal sensor, and profile data, waviness data, roughness data, $Ra$, and $Rz$ parameters are generated in the developed data-processing software. The assembled measuring system based on the chromatic confocal laser sensor showed its performance in assessing the roughness parameter $Ra$, from 0.34 μm to more than 12 μm, which covers a common range of milling, turning, and grinding. In this range, measurement relative errors can be controlled within 10%. Frequency analysis and correlation analysis of profilograms were performed. Frequency analysis made it possible to establish the dominant frequency components that occur in the profilogram of the samples, while correlation analysis was used to develop a methodology for identifying the deterministic and random components of the processed surface profile signal. The results of the analysis can be further used to develop diagnostic functions for process monitoring based on profilogram estimates, such as the autocorrelation function and the power spectrum density.

**Keywords:** chromatic confocal sensor; non-contact measurement; surface roughness; autocorrelation function; power spectrum density





## 1. Introduction

The quality of the machined surface, along with the required physical and mechanical state of the surface layer and the accuracy of machining, is the most important complex indicator that determines the performance properties of machine parts. The surface roughness of machine parts and devices, which characterize the surface quality, has a significant impact on the operational characteristics of machine parts and products. Advanced manufacturing technology requires in situ surface finish measurement. This question is relevant both for parts of a simple form, and for free-form parts, 3D-printed products [1,2], and mirrors [3]. Mounting and dismounting of the part leads to systematic errors and geometric deviations, so it is necessary to be able to estimate the surface roughness parameters without removing the part from the machine.

Along with the contact method of measuring roughness, optical non-contact methods are widely used, each of which has its own optimal field of application. The confocal chromatic sensor, having such undoubted advantages as working with various types of materials, high repeatability, etc., requires research to identify the range of surface roughness without noise, taking into account the effect of the scanning speed. Assuming that the contact method is a reference method, it is necessary to determine the error of the non-contact method compared to the contact method on a specimen and metal samples having two types of profiles: periodic and aperiodic.

Functions such as the autocorrelation function and the power spectral density provide more information about the surface and the process that formed the profile in the time and

frequency domains, respectively. To analyze these types of profiles, it is possible to apply the technique presented in this paper for dividing the initial profile into a deterministic and random component.

The aim of the research in this area is to develop a measurement system based on the chromatic confocal sensor, which should be open to the possibility of further improvement according to specific conditions of use, especially in the field of software. The system can be useful in research, teaching, business, and manufacturing. After the approbation of the measuring system, and after appropriate modernization, it can be integrated into the technological process of parts' machining for in situ surface quality control.

In this paper, a non-contact surface finish measuring system is proposed, and the paper is organized as follows: Section 2 shows the existing work in this area and substantiates the relevance of the research, and Sections 3.1 and 3.2 present the materials and methods used in the study. Section 4.1 compares stylus-based and confocal-based methods for surface roughness measurement of the standard specimen, Section 4.2 presents the proposed system configuration and validation for the surface roughness measurement of the samples, analyzes the experimental work and measurement data, and presents the determination of the scope of the developed system, and Section 4.3 discusses the autocorrelation function and the power spectrum density in terms of their potential use for process monitoring. Finally, Section 5 presents the conclusions.

## 2. Literature Review

Since two main components are involved in the production of any workpiece: the production process and the machine or production technique (Figure 1), it is obvious that the surface finish will be sensitive to any changes in the process. Surface finish is one of the parameters that determine the surface quality and limit the productivity of the technological operation. This parameter is important in terms of friction, contact, wear, lubrication, reflectivity, and other characteristics. Therefore, it is logical to assume that surface measurement can be used to control the manufacturing process. Moreover, if the controlled surface parameter remains constant from workpiece to workpiece, then the process has the property of stability. When the observed output parameter of the technological system deviates, it is necessary to change the parameters of the system state, which will reduce the production costs.

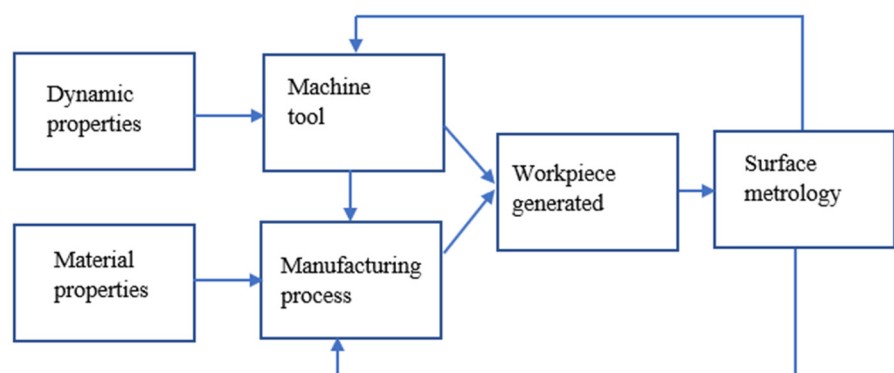

**Figure 1.** Surface measurement and manufacturing [4].

In this case, the measurement of the workpiece must be carried out at the workplace without removing the workpiece from the fixture. Two methods are currently used in production: contact and non-contact surface roughness measurement methods.

The generally accepted contact method for surface quality control, the so-called stylus profilometry (contact stylus surface scanning), is as follows. The diamond needle is pressed and moved parallel to the surface being examined. In places with irregularities (peaks and valleys), mechanical vibrations of the measuring head of the needle occur. These vibrations are transmitted to a sensor that converts the mechanical energy of the vibration into an

electrical signal, which is amplified by the transducer and measured. Since the needle tip is in contact with the surface, the roughness profile is a copy of the surface [5]. However, if the needle comes into contact with the surface, the latter may be damaged by the probe. The needle is worn during the measurement process, and the measurement is limited by the radius of the stylus tip. Measurement results depend on the stylus wear and radius. This method is not used when measuring viscous samples [6]. The contact method is quite slow and has a flanking error [5]. All the above limitations prevent the use of this method for in situ surface roughness measurements. Non-contact methods have been developed to avoid the disadvantages of contact methods.

Among non-contact measurement methods, optical methods such as non-contact laser technology, confocal microscopy, and interferometry have become widespread. Improvements in the manufacturing technology of laser sensors have led to their widespread is use in laboratory and real production conditions. Non-contact laser technology is used for displacement, distance, and position measurements [7]. Among the most common laser sensors, there are two types: the triangulation sensor and the confocal sensor, which differ in their principles of operation. The principle of operation of the laser sensor is based on the triangulation method for measuring the distance to an object. The triangulation method uses the angle of the reflected beam to determine the distance to the object. The angle of incidence of the laser beam changes depending on the distance to the object, and thus the position of the laser point on the receiver, which is a photodiode line, changes. The light-receiving element consists of a set of light sensors that can determine the position of the received beam [8].

A diagram of a typical design of a confocal head is shown in Figure 2. Only one selected frequency from the beam incident on the surface is focused on the passive optical system, which depends on the height of each point and provides a clear image of this point by one photodetector. The photodetector is an accurate spectrometer that allows to determine the wavelength that provides information about the height of the measurement roughness [3,9].

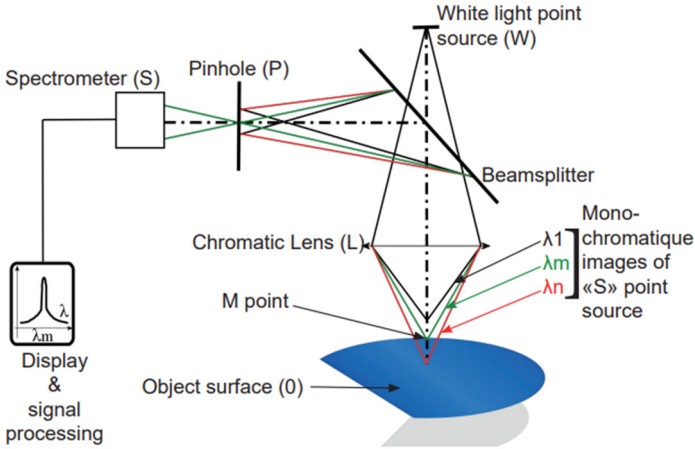

**Figure 2.** Principle work of the confocal sensor [10].

Surface information with confocal microscopy is generated by acquiring successive confocal images through the depth-of-field of the objectives. Objectives with a higher numerical aperture can reduce noise, but due to their small fields of measurement, in most cases a combination of several areas (stitching) is required, which requires additional time. On the other hand, lower-magnification lenses have a comparatively larger measurement area but have lower vertical resolution and less perception of tilt, which means surfaces with localized steepness are difficult to measure [11]. First, the samples must be small enough to fit into the microscope cell. Second, samples must be prepared for electron microscopes, since non-conductors cannot be directly measured [12].

Coherent scanning interferometer (CSI) refers to a class of optical surface measurement methods wherein the localization of interference fringes during a scan of the optical path length provides a means to determine surface characteristics such as topography, transparent film structure, and optical properties. Both the shape of the interference fringes and their variation in contrast are suggestive of the surface profile.

However, these instruments are too expensive, and need strict experiment conditions; therefore, they are mainly used for sample inspection and studies in a laboratory. Interferometry methods can also obtain a high accuracy, and can realize 3D topography measurements; however, the structures of the experiment setup are usually very complex, and the operation is not simple and convenient, so they are limited in their applications [12]. Optical methods also have their drawbacks compared to, for example, tactile sensing. Possible problems include sensitivity to reflectance, changes in the specular and scattered light intensity due to the surface treatment or tilt, and multiple reflection and distribution paths in the sample [13].

A large number of papers are devoted to the comparison of contact and non-contact measurement methods. Such a need most often arises due to the disadvantages of contact methods and the study of the possibilities of a wide implementation for production. However, since the contact method is a reference method, the question arises of conducting appropriate studies.

Wagner et al. [14] compared the roughness measurement of gear teeth performed using optical interferometry with stylus profilometry. Recommendations for choosing an interferometer objective lens and a scan length were provided. It was noted that the optical interferometry measurement was more time-consuming than the stylus profilometer measurement. Moreover, additional time was required for the production of replica samples. The results obtained by the two methods were close, but the readings obtained using the optical interferometry were lower.

Poon and Bhushan [15] performed comparative measurements of the roughness of a glass-ceramic substrate with the stylus profiler (SP), atomic force microscope (AFM), and non-contact optical profiler (NOP). The AFM method provided the most accurate roughness measurement since the glass-ceramic substrate measurement has submicron wavelength structures that will not be detected by NOP and SP, with a 0.5 μm tip.

Mital' et al. [16,17] compared the roughness parameters, *Ra* and *Rz*, measured on metal samples with a Mitutoyo SJ-400 contact profilometer and a triangulation laser profilometer. The measurement assembly consists of a laser beam source, the lens, and a CMOS sensor camera. The *Ra* values measured by the laser profilometer were in the range of values measured by the contact method, and the *Rz* values were 2–3 times less than the values measured by the laser due to the peculiarity of the stylus contact with the measured surface, and partly due to the noise in the laser measurements. Sometimes, sharp peaks appear in the results at certain laser settings. These distorted values must be manually removed from tables in the file. The obvious advantage of laser profilometry is the formation of a complete pattern of the estimated surface during automatic movement of the head, i.e., the formation of a set of values describing the surface area at each measurement step.

The beam width (measuring range) limited the measured area's width. To obtain a correct result, it becomes necessary to carefully adjust the operating mode of the profilometer, which is characterized by the absence of noise visible in the profile image. The influence of the reflectivity of the material used can introduce an error in the measurement result. At the same time, the different initial qualities of the studied surfaces require certain settings of the laser profilometer to obtain the correct result.

Fu et al. [5] presented the developed chromatic confocal system integrated with a 7-axis industrial robot arm for the in situ surface roughness measurement. A commercial chromatic confocal sensor was selected as the sensor. Based on the wavelength of the reflected light, the position of the focal point can be measured from the detected wavelength. A precision reference specimen (178-602, Mitutoyo Corporation, Takatsu-ku, Kawasaki, Kanagawa, Japan) found that the chromatic confocal measured profile had higher mea-

surement distortion and spikes (peaks and valleys). Next, the parameters *Ra* and *Rq* were measured by the commercial stylus profilometer Talysurf and a non-contact method on 3D-printed, curved blades. In this case, the relative error for both *Ra* and *Rq* parameters was less than 5%.

The coherence scanning interferometry and confocal profilometry methods were compared with the tactile method for a 3D-printed object [1]. Studies have shown the ability of the coherent scanning interferometer to scan steep grooves, as well as the consistency of *Ra*. The laser confocal sensor Keyence LT9000 was used as a laser sensor for the single-point height measurement. The measured profile was superimposed on the profile of the coherent scanning interferometer (CSI) (chosen as the reference profile). The author noted the expediency of using the profile obtained by coherent scanning interferometry as a reference, especially when the surface finish was low, and especially for the *Rz* parameter. The use of the laser confocal sensor showed close results to those of the reference method.

A measuring system based on the Keyence LT-9010M laser confocal sensor, which is integrated into a robotic arm, was proposed [18]. The data confirmed this system's ability to measure surface roughness in the *Ra* range of 0.2 to 7 μm, with a relative accuracy of 5% compared to the contact method using the Talysurf PGI 800 stylus profilometer. The measured speed can reach up to 3 mm/s compared to the stylus travel speed of 1 mm/s. The author initially tested the performance of the sensor on a calibrated specimen (*Ra* 2.97 μm, Mitutoyo Corporation, Takatsu-ku, Kawasaki, Kanagawa, Japan). The measurement length reached by the probe was 1.1 mm, which covers only one cutoff length of 0.8 mm. In view of this limitation, a data-stitching algorithm was introduced to eliminate the jump errors observed on overlapping measured profiles. Due to the specified algorithm, it was possible to achieve a surface profile measurement of up to 12.7 mm in a trace length. Next, a set of roughness samples with a nominal value of *Ra* in the range of 0.2 to 6.3 μm was used to calibrate the measuring system. For very smooth surfaces with *Ra* values less than 0.4 μm, the laser confocal system showed unpredictable errors due to two typical disadvantages: the spot size limit and the background noise. The following disadvantages of the measuring system were revealed:

1. Samples are recommended to be measured at certain settings for cutoff, evaluation length, and stylus travel.
2. The measuring system has a trace length limit of 12.7 mm.
3. To obtain a profile of the measured surface, it is necessary to use the data-stitching algorithm, since they are limited in length to 0.8 mm.
4. It is not indicated whether the profile type (periodic, non-periodic) was taken into account when choosing the settings and whether the profile type affects the relative measurement error.

An experimental work consisting of measuring the surface finish with the confocal displacement sensor based on the chromatic imaging principle system was performed [19]. A set of standard machined surfaces (turned and vertical-milled) have been researched. It was observed that the measured *Ra* values using a confocal sensor and the standard *Ra* value showed significantly less variation compared to the stylus method for the machined surfaces. Significant relative errors were revealed when using the confocal displacement sensor. For example, for the turning sample, the error varied by up to 37%.

The confocal chromatic sensors allow a tilt angle of up to ±30° and have a high numerical aperture. This provides high resolution and small light spots. This makes it possible to accurately and reliably detect curved and structured surfaces.

Due to the non-contact measurement principle, the sensor does not affect the target object, which allows wear-free measurements [20]. In confined spaces, or where high reflective surfaces or transparent objects need to be measured, confocal chromatic sensors are capable of measuring the thickness of transparent materials [21]. In terms of the measurement time, this is the only method that allows to obtain the thickness of the workpiece in one measurement. The sensors are capable of sizing in both single- and multi-layer glass applications.

Whether rough or specular, the system can handle a wide range of materials, including glass, stainless steel, and ceramic. Confocal chromatic sensors benefit from the fact that the beam path is aligned with the axis of the sensor, which eliminates the shadowing effect seen in other optical distance measurement methods with displaced light transmission and reception points, such as laser triangulation. This allows confocal chromatic sensors to measure inside recesses without a loss of accuracy.

Submicron- or non-scale measurement accuracy is achieved because this technology allows each point to be positioned in the axial direction in accordance with the light of different wavelengths. It has good repeatability and a short measurement time, for example, compared to the image measurement method [22]. The author of [22] notes that such methods as optical fibers and atomic force microscopes are unsatisfactory in accuracy, affordability, and efficiency.

Beyond distance and thickness measurements, confocal chromatic sensors offer a unique capability to generate detailed surface topographies by tracking the intensity of the signal as the sensor scans across an object. This process shows even the finest surface structures, revealing flaws such as scratches or a damaged finish. Material differences are also highlighted due to changes in reflectivity as the sensor passes over various substances in the target.

Among the advantages of the chromatic confocal system compared to the triangulation laser profilometer system is the ability to precisely measure on all targets, including transparent, mirrored, unfinished metal, ceramic, and adhesive surfaces, and to stably measure on targets that cast multiple reflections or absorb light, effective on curved, uneven, and rough surfaces [23]. The chromatic confocal sensor uses a white light LED, such as commercial chromatic confocal sensors [5,10], or an LPD light source, which are available in the CL-3000 series sensors, Keyence. They differ in the advantages of the LPD light source [23] compared to LED. Using an ultra-high-brightness, multi-color transmitter LPD light source allows for a larger measurement range and higher accuracy across the entire measurement range, compared with confocal displacement sensors using white LED light sources [23].

At the same time, LED sources are used in the non-contact confocal 3D profilometer [24] and the line scanning confocal microscope [25,26].

An urgent issue in the study of confocal chromatic probe sensing is the study of the speed effects [27], and the state of the measured surface in relation to outliers in the results of surface measurements after micro-EDM. It was found that at scanning speeds of 1 mm/s and 10 mm/s, large spikes were found on the height maps of irregularities [27]. These spikes are considered outliers since their amplitudes are higher than the neighboring points. Among the reasons for this are false reflections of light, the transparency of the material, and the local curvature, and at a higher speed, there are more spikes.

To overcome these shortcomings, one can apply interpolation for unmeasured points at a high scan rate based on neighboring points and filter out the outliers using median and morphological filters. Surface contamination can make additional distortions into the measured roughness parameters, manifesting itself in their increase.

Ye et al. [28] explored the uncertainty of measuring area texture parameters measured by a chromatic confocal microscope built into an EDM machine in a workshop environment, where vibrations and temperatures must be taken into account. For example, machine vibration can increase the surface roughness by 10%. The author of [29] investigated the effect of the measurement speed with a confocal chromatic sensor on the areal roughness. Recommendations were formulated on the settings for measuring the parameters Sq, Sa, and Sz. Therefore, the study of the influence of speed on the quality of the collected data is a topical issue.

As noted above, the surface measurement can be used to diagnose the state of the technological system and control the production process. In other words, if the measured parameter, the calculated parameter, or the function characterizing the surface is constant from workpiece to workpiece, then the technological process is stable. It is noted that the

use of a number of simple parameters does not provide the designation (identification) of changes in the production process [4]. The emergence of the theory of random processes is a diagnostic tool that allows developing evaluation functions—diagnostic features for online monitoring of surface quality. As diagnostic features for monitoring the surface quality of a 3D-printed plastic object, a set of parameters was analyzed: RMS, standard deviation, variance, median, mode, summation, and arithmetic mean, found from the power spectrum density [2]. The least and most sensitive information signals for monitoring defects in a 3D-printed object have been identified.

The use of functions such as the autocorrelation and power spectra can reveal much more information about the process [4]. They are more reliable, since in the case of auto-correlation and power spectrum density, any random phase shifts between the sinusoidal components that make up the profile are eliminated. According to Whitehouse [4], the autocorrelation function is informative for observing random surfaces, and the power spectrum density is informative for periodic ones.

Aich and Banerjee [30] proposed a non-dimensional index PR ratio, found using an autocorrelation function, to estimate the relative contribution of randomness and periodicity in the surface topography. The PR ratio is the ratio of the segment lengths corresponding to periodicity and randomness on the correlogram. The autocorrelation function can be used to validate an area from the removal methods and determine high-frequency shape errors [31]. Errors in the calculation of roughness parameters can be reduced by applying the autocorrelation function.

The main prerequisites for the widespread use of laser technologies in engineering applications for measuring surface roughness are that they are non-contact, non-destructive, and relatively fast. With the development of computational capabilities, these methods quickly provide surface information, with a high resolution. The potential of these methods lies in the possibility of their operational use.

Almost all researchers note the prospect of integrating a measuring system based on a laser sensor into the technological process of manufacturing a part for in situ quality control. The authors compared contact and non-contact methods based on triangulation and confocal sensors, thus determining the possibilities and conditions for their application. Exploratory studies of the conditions of use are due to the drawbacks of the non-contact method. For example, changes in reflectance and light scattering can result in image noise or dropouts. A significant vertical measurement range is required to capture surface vibrations over a large scan range. The authors found the measurement error characterizing the difference in the readings obtained by the two methods. Moreover, the error and scope must be determined for each type of sensor built into the measuring system.

Additionally, there is no study of smooth surfaces with *Ra* less than 0.2 μm in the literature. It would be interesting to compare the results of roughness measurements with different confocal sensor models, which differ in the laser spot size and resolution when measuring metal samples, with a wide range of roughness *Ra* of 0.05 μm to 12.5 μm. A necessary task is to study the effect of speed on the measurement uncertainty for the selected range of changes in the roughness parameters of the metal samples.

The next point is that, since all surfaces obtained as a result of mechanical and abrasive processing can be attributed to periodic or aperiodic profiles [32], it is promising to study the evaluation functions (autocorrelation function and power spectrum density) for these types of profiles in order to further use them in online process monitoring.

In addition, it is possible to consider the autocorrelation function and power spectrum density for the same samples from a unified standpoint using the methodology for identifying the ratio of deterministic and random components based on the dispersions of these components, as well as identify the distribution of the specified ratio for samples with different nominal roughness, but processed by the same mechanical method.

A measuring system for the non-contact measurement and evaluation of the surface roughness of parts based on the chromatic confocal sensor CL-PT010 has been developed

at the Department of Mechanical, Manufacturing, and Biomedical Engineering of Trinity College, Dublin. Therefore, the question of studying this measuring system arose.

The objectives of the current study are as follows:

- Carry out a comparative analysis of the results of contact and non-contact methods for measuring surface irregularities of a metal specimen with a known process roughness. Set the relative error of the non-contact method compared to the contact method.
- Conduct comparative studies on metal samples—metal surface roughness standard set, obtained by mechanical processing and with two types of profiles: periodic and non-periodic. Set the relative error of the non-contact method compared to the contact method in a wide range of surface roughness parameters.
- Investigate the influence of speed on the measurement uncertainty for the selected range of the roughness parameter under study, taking into account the noise arising from the drive unit and the sample state.
- Set the roughness range of surfaces obtained by machining methods, taking into account the possible noise from the surface gradient, the drive noise, etc.
- Identify the scope and the settings of the chromatic confocal laser sensor for the purpose of determining the roughness parameters.
- Study the profilogram as a signal in the time and frequency domains, which will provide additional functions for monitoring the process (operation). Study the autocorrelation function and the power spectrum density for periodic and aperiodic profiles.

## 3. Materials and Methods

### 3.1. Stylus Measurement

The stylus tip physically senses the sample surface, traversing across the surface with a static measuring force of 0.75 mN or 4 mN, depending on the Stylus tip according to ISO 3274 [33]. The technical parameters of the contact roughness tester are shown in Table 1.

**Table 1.** Specification of SJ-400.

| Parameter | Value |
|---|---|
| Measuring range Z-axis | 800 μm, 80 μm, 8 μm |
| X-axis | 25 mm |
| Cutoff length | 0.08, 0.25, 0.8, 2.5, 8 mm |
| Measuring speed | 0.05, 0.1, 0.5, 1 mm/s |
| Measuring force | 0.75 mN |
| Minimum resolution | 0.000125 μm (8 μm range) |
| Arbitrary length | 0.1 to 25 mm |
| Digital filter | 2CR, PC75, Gauss |

To compare the measured values of the surface roughness parameters of the samples, the roughness measurement was carried out on the commercial stylus profilometer Portable Surface Roughness Tester Surftest SJ-400 (Figure 3). To obtain the roughness parameters via two methods, a precision reference roughness sample (Mitutoyo 178-602, Mitutoyo Corporation, Japan) with the parameter *Ra* value of 2.97 μm (Figure 4) was used.

### 3.2. Laser Confocal Measurement Method

The SR-Test Precision for the non-contact measurement and evaluation of the surface roughness of parts based on the chromatic confocal sensor has been developed at Trinity College, Dublin. In this study, a single-point chromatic confocal sensor (Table 2), Keyence CL-PT010 (KEYENCE, Osaka, Japan), was used for developing an in situ roughness measurement system.

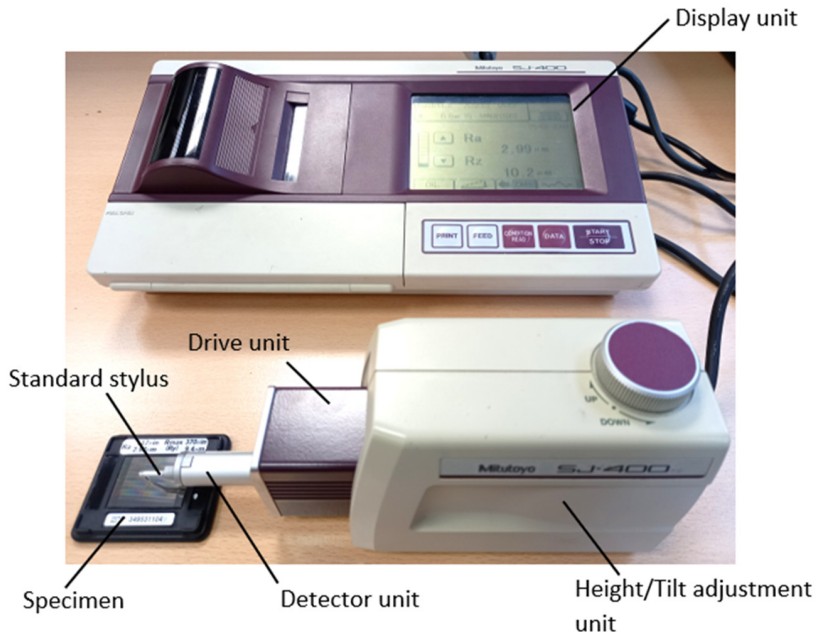

**Figure 3.** Contact method—Mitutoyo SJ-400.

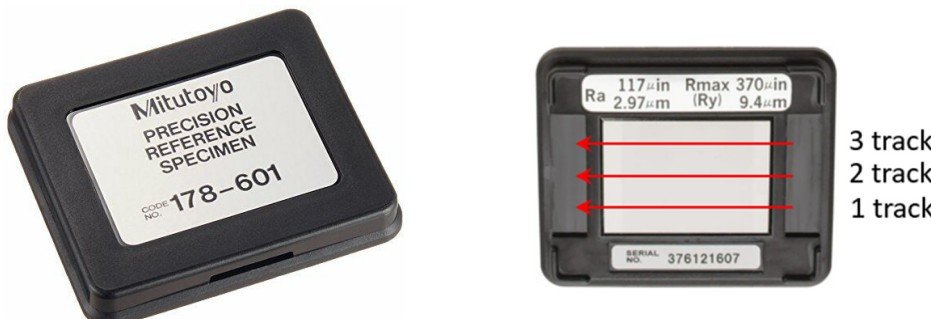

**Figure 4.** Precision roughness reference specimen with the measuring direction.

**Table 2.** Sensor head CL-PT010 specification.

| Specification | | Value |
|---|---|---|
| Reference distance | | 10 mm |
| Reference measurement range | Measurement range | ±0.3 mm |
| | Linearity | ±0.22 μm |
| High-precision measurement range | Measurement range | ±0.15 mm |
| | Linearity | |
| Resolution | | 0.25 μm |
| Spot diameter | | 3.5 μm |

The sensor heads of the confocal displacement sensor CL-3000 series [23] are extremely compact and lightweight compared to conventional heads. They are also unaffected by thermal or electrical noise, so they can be installed inside equipment even in confined spaces, providing high measurement accuracy and stability. In addition, the CL-3000 series can accurately distinguish between the top and bottom surfaces of transparent objects with a thickness of 15 μm or more (such as glass substrates, transparent films, and thin layers of transparent materials), allowing height (distance) measurement [34].

SR-Test Precision is a standalone benchtop machine intended for use in a laboratory environment to perform surface displacement and roughness measurements. Its operating principle is to gather 2D line scans of a part by moving the part underneath the part of

a chromatic confocal sensor. This is achieved by placing the part in an automated XY stage. The sensors are mounted to an automated Z-axis, which is positioned prior to measurements so that the surface of the part is within the range of the sensor(s) being used. The Keyence chromatic confocal system consists of three main hardware components: the CL-PT010 optical probe, the CL-3000 controller, and the optical unit (Figure 5). The last one contains the light source unit, Quad CMOS, and the spectrometer. Multi-color light is generated using an LPD light source that simultaneously emits red and green light. The emitted light is more stable and of a higher brightness over a wider range of wavelength bands compared to typical white LEDs [23]. The Keyence chromatic confocal system is setup as shown in Figures 6 and 7. The chromatic confocal sensor is connected to the optical unit using the fiber cables. The optical unit is directly linked to the CL-3000 controller via a connector and fixed in place using screws. All control/communication to/from the chromatic confocal sensor goes through the controller, which has USB, ethernet, and RS-232 ports for PC communication, and DI and DO terminals for control inputs and readouts using lead wires. The linear stage has a travel range of 40 mm. The measuring system generates files in the .cvs format, which are imported into MATLAB (2021), where, when working with scripts, the user generates the measured profile estimates, namely the *Ra* and *Rz* parameters according to ISO 4287 [35].

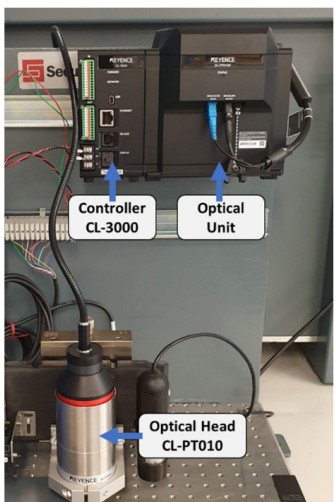

**Figure 5.** The Keyence chromatic confocal system.

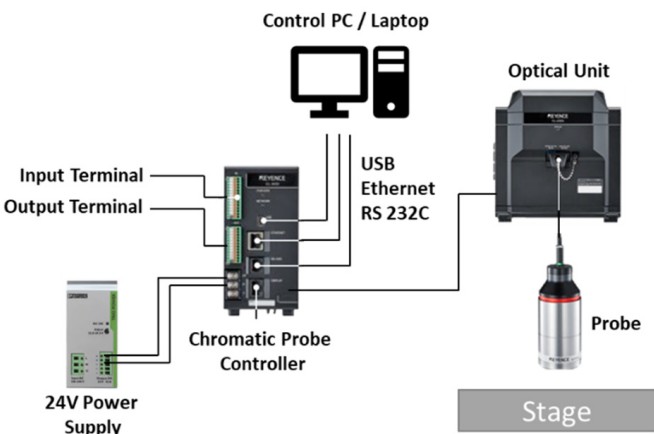

**Figure 6.** Chromatic confocal sensor wiring diagram.

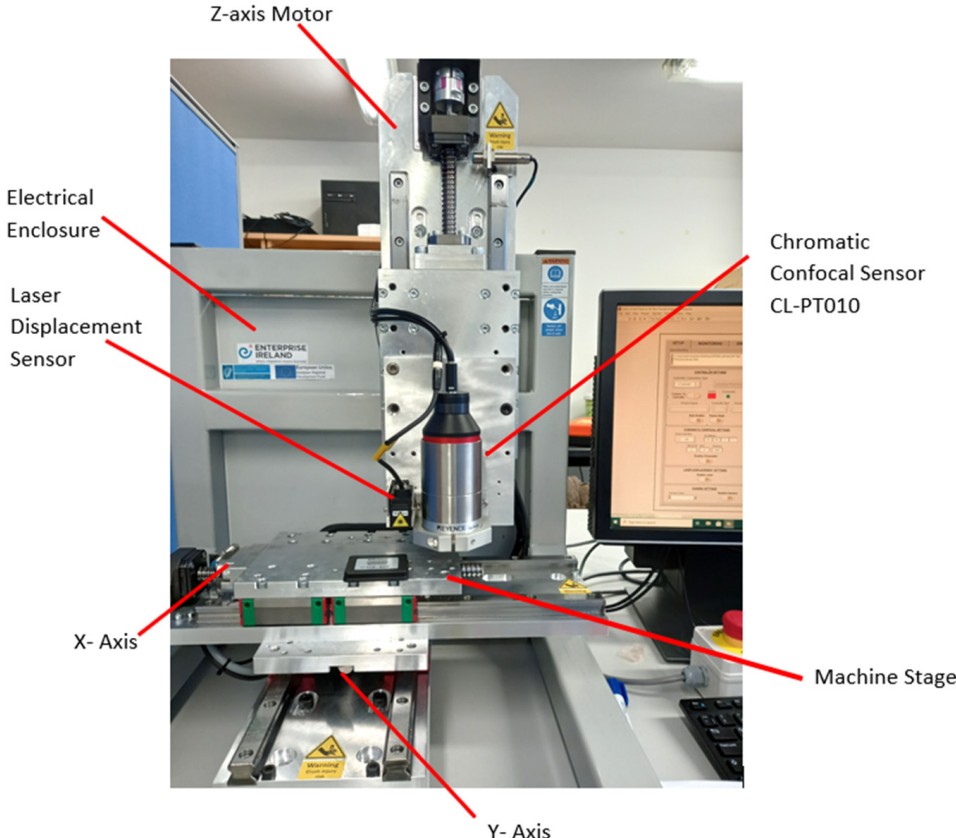

**Figure 7.** Experimental setup during measurement using CL-PT010 of the roughness reference specimen.

The sampling frequency of the laser confocal sensor was able to achieve 1 kHz, which is suitable for surface profile measurements. A 2D surface profile was processed in real time by an algorithm to compute the surface roughness parameters.

## 4. Results

### 4.1. Comparison of Stylus and Confocal Measurement Methods

A comparison of the roughness parameters obtained by two methods: the contact method, based on the stylus profilometer Portable Surface Roughness Tester Surftest SJ-400, and the non-contact method, based on the assembled measuring system with the chromatic confocal sensor, was performed. Roughness parameters were determined on a standard metal specimen (*Ra* 2.97) and a surface roughness comparator standards composite set (nominal roughness: 0.05 µm to 12.5 µm).

Sample Mitutoyo 178-602, Mitutoyo Corporation, Japan, was measured three times on each of the three tracks according to Figure 4. Measurement conditions were: the evaluation length *ln* was 12.5 mm, the sampling length *lr* and the cutoff *λc* were 2.5 mm, the traverse length *lt* was 15 mm, the traversing speed was 0.5 mm/s, the measuring range was 80 µm, and the number of sampling lengths was 5. The results of measuring the roughness parameters: *Ra*, *Rz*, sample standard deviation, and margin of error, for SJ-400 (Figure 3) and CL-PT010 (Figure 7) are presented in Tables A1 and A2 (Appendix A), respectively.

The profilogram of the specimen surface had a harmonic form (Figure 8). For example, at *V* = 0.05 mm/s, number of tests = 1, and number of tracks = 1, the obtained parameters were *Ra* 2.98 µm and *Rz* 9.50 µm using SJ-400 (Table A1, Appendix A) and *Ra* 3.02 µm and *Rz* 10.52 µm using CL-PT010 (Figure 8, Table A2, Appendix A).

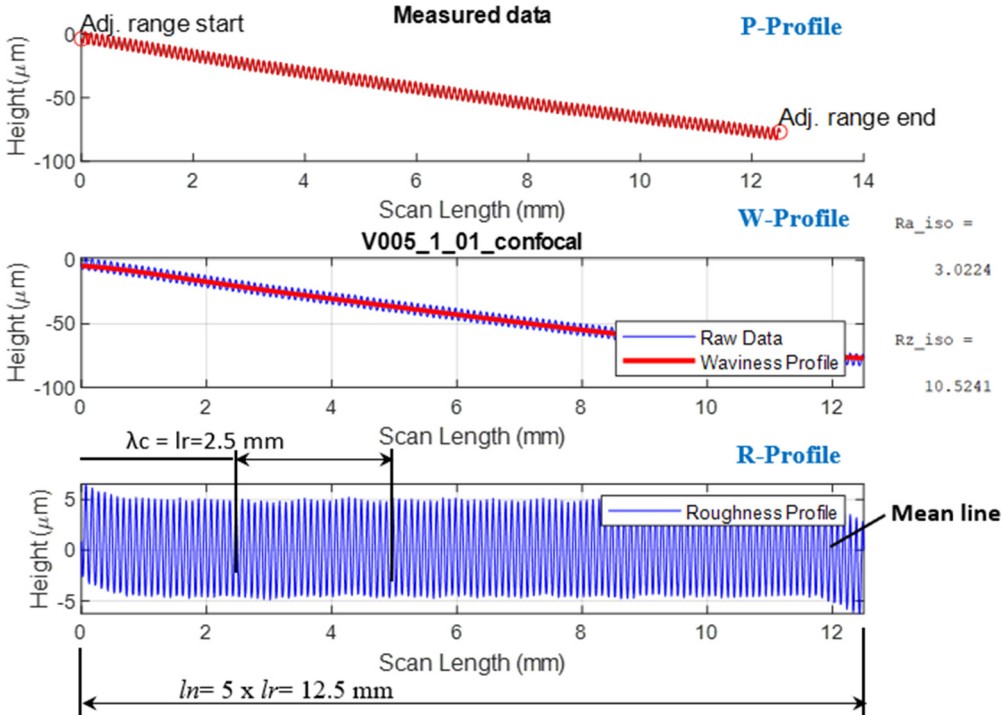

**Figure 8.** Screenshot of the MATLAB program for processing the measurement results.

The following conclusions were drawn from Table A1 (Appendix A):

1.  When changing the stylus speed, the sample standard deviation of the *Ra* parameter varied from 0 to 0.012 μm, while the *Rz* parameter varied from 0 to 0.462 μm. Moreover, the largest values referred to the maximum probe movement speed of 1 mm/s.
2.  The margin of error for the *Ra* parameter varied from 0 to 0.014 μm, and for the *Rz* parameter it varied from 0 to 1.147 μm. Moreover, the largest value referred to the maximum probe movement speed of 1 mm/s.
3.  When the probe movement speed increased from $V = 0.05$ mm/s to $V = 1$ mm/s, the sample standard deviation and margin of error parameters for the *Ra* parameter remained constant, while for the *Rz* parameter they were variable values. In this case, the average *Ra* values were averaged over three measurements of the roughness parameters measured at the same place in each track, then averaged over three tracks.
4.  The probe movement speed had little effect on the parameters *Ra* and *Rz*. For example, when the probe speed increased from 0.05 mm/s to 1 mm/s, the parameter *Ra* changed by 0.43%, and *Rz* by 0.2%. Therefore, it is possible to set the maximum speed of the probe when measuring.

Table A2 (Appendix A) showed the following:

1.  When the speed of the measuring head increased, the average values of the parameters *Ra* and *Rz* decreased. In this case, the average values were averaged over three measurements of the roughness parameters measured at the same place in each track, then averaged over three tracks.
2.  When the speed of the measuring head increased from $V = 0.05$ mm/s to $V = 0.15$ mm/s, the sample standard deviation and margin of error for the *Rz* were less than those for the *Ra* parameter. When the speed of the probe increased from $V = 0.20$ mm/s to $V = 1$ mm/s, the opposite trend was observed.
3.  When changing the speed of the measuring head, the sample standard deviation of *Ra* varied from 0 to 0.145 μm, while for *Rz* from 0.013 to 0.939 μm.
4.  The margin of error for *Ra* varied from 0.002 to 0.181 μm, and that for *Rz* from 0.031 to 0.707 μm.

The relation of the parameters *Ra* and *Rz* (averaged values over three tests on three tracks) on the probe movement speed had an exponential decreasing character (Figure 9a,b, respectively). From Figure 10, it can be seen that with an increase in the speed of the measuring head, the error in the estimates of *Ra* and *Rz* increased. An error level of 5% corresponds to the probe movement speed of *V* = 0.2 mm/s.

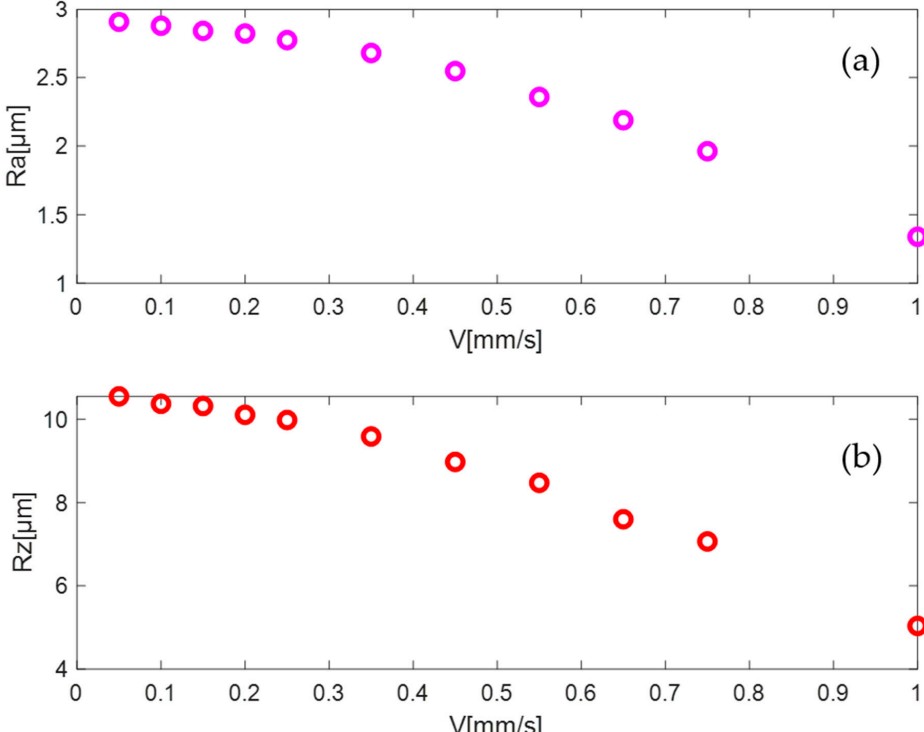

**Figure 9.** The parameters *Ra* (**a**) and *Rz* (**b**) vs. the speed of the measuring head.

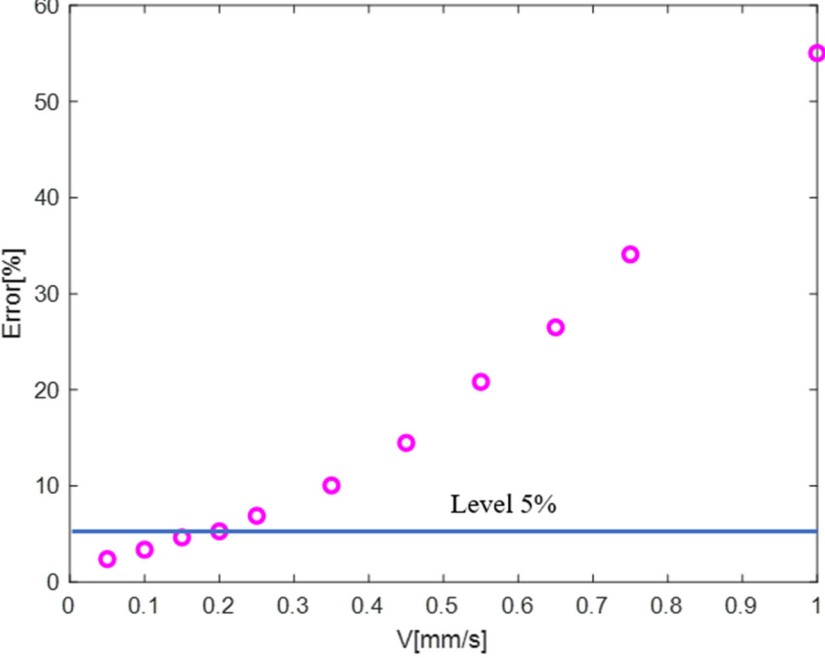

**Figure 10.** Relative error vs. laser sensor speed.

### 4.2. Roughness Measurement of Sample Roughness Set

To determine the accuracy of the proposed confocal system, a profilometer with an SJ-400 probe was used as a reference. Roughness standard samples (Figure 11) obtained by various mechanical and abrasive methods with a nominal *Ra* value in the range of 0.05 to 12.5 μm were used to evaluate the measurement accuracy of the chromatic confocal sensor.

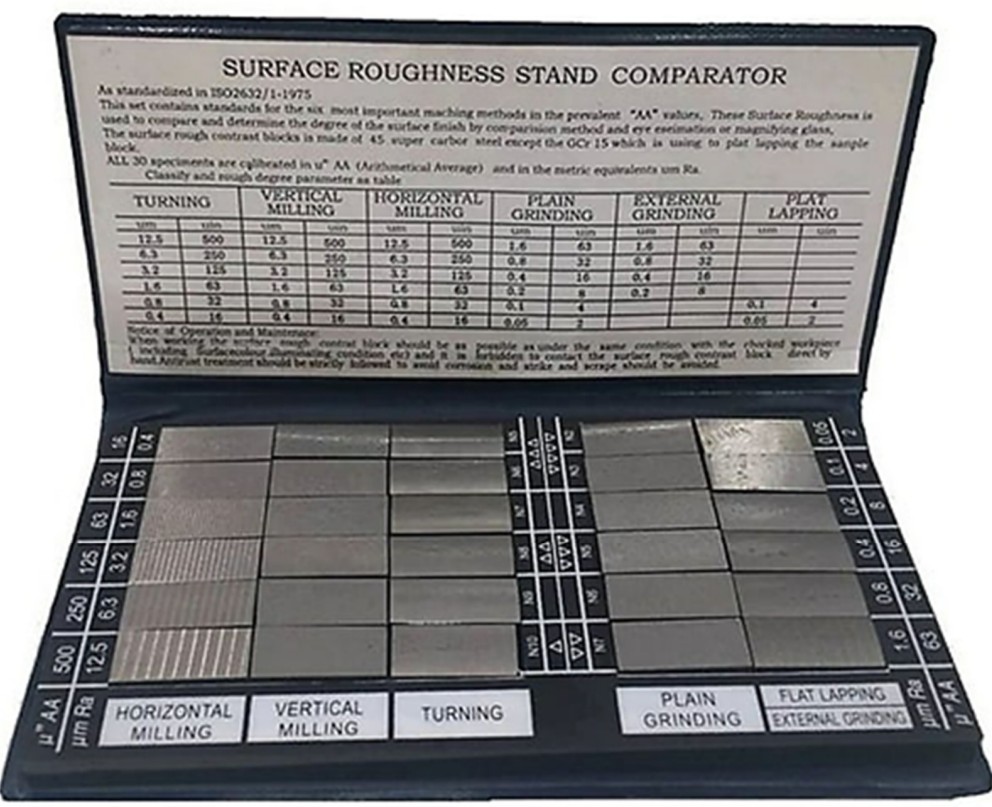

**Figure 11.** Surface roughness comparator standards composite set.

The DIN EN ISO 4288 standard [36,37], in accordance with which the SJ-400 station operates, provides for a cutoff equal to the base length, to be selected while taking into account the type of measured profile: periodic or aperiodic profile (Figure 12). The selected roughness samples are classified depending on the type of profile: periodic profiles—turning, vertical, and horizontal milling, and aperiodic profiles—plane, external grinding, and lapping.

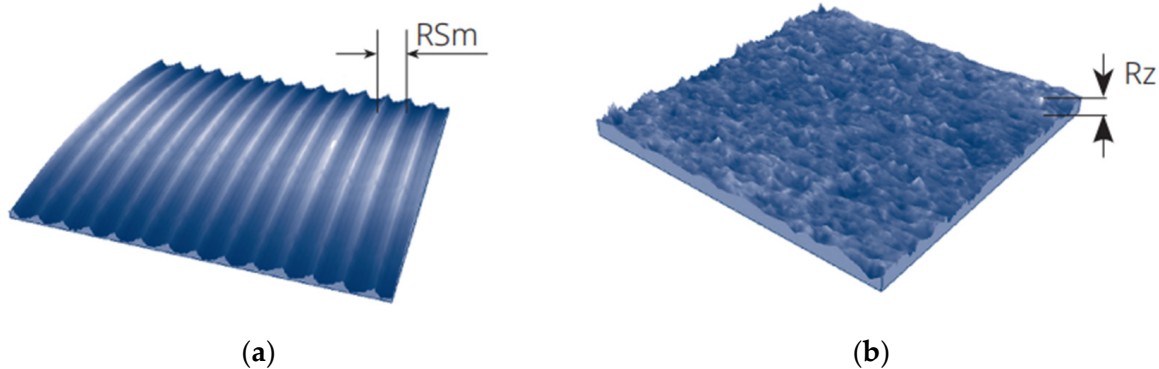

(**a**)　　　　　　　　　　　　　　　　　　　　　　　(**b**)

**Figure 12.** (**a**) Periodic and (**b**) aperiodic profiles in the topography of the roughness of the machined surface [32]: *RSm* and *Rz* step and height parameters of the profile.

All samples were measured by contact and non-contact methods (Figure 13) with the following settings: evaluation length *ln* = 12.5 mm, cutoff *λc* = 2.5 mm, stylus travel

$lt$ = 15 mm, traversing speed = 0.5 mm/s, measuring range = 800 µm, and the number of sampling lengths = 5. For the aperiodic profile, the values of the roughness parameter $Ra$ were compared with the table in [32] and, in accordance with the recommendations, the tuning parameters $\lambda c$, $ln$, and $lt$ were adjusted. For periodic profiles, the value of the roughness step was found from the profilograms. In accordance with the step, the parameters $\lambda c$, $ln$, and $lt$ were corrected according to the recommendations [32], and the measurements were repeated. The $Ra$ parameter was evaluated by measuring three tracks with a distance of 3 mm between them. Each track was run three times, with the stylus and the chromatic confocal laser sensor moving at 0.05 mm/s.

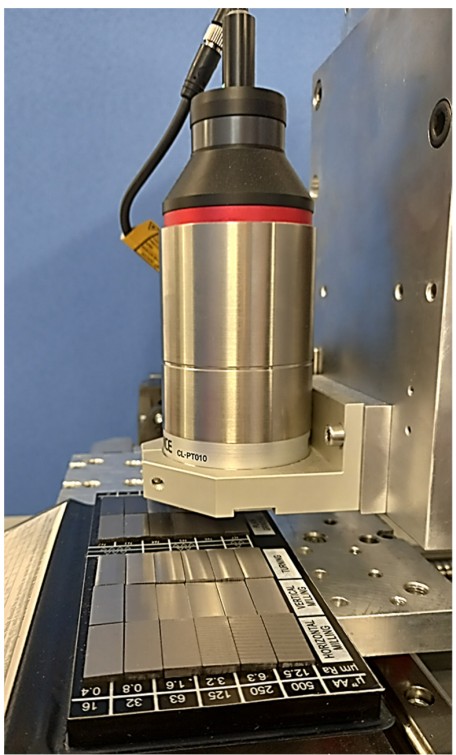

**Figure 13.** SR-Test Precision during the measurement of roughness standard samples.

The measurement errors of the chromatic confocal system in Table 3 are plotted as the error in Figure 14.

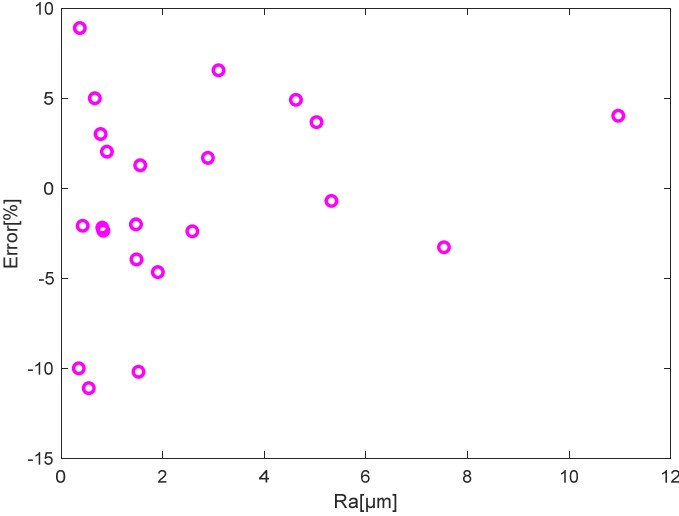

**Figure 14.** Relative error vs. *Ra*.

**Table 3.** *Ra* of different surfaces measured using a confocal system and a stylus profilometer.

| Machining Method | *Ra*, μm | SJ-400, Contact Method | Chromatic Confocal Sensor, Non-Contact Method | Error, % |
|---|---|---|---|---|
| Vertical milling | 12.5 | 10.97 | 11.41 | 4.04 |
| | 6.3 | 4.62 | 4.85 | 4.92 |
| | 3.2 | 2.89 | 2.94 | 1.70 |
| | 1.6 | 1.47 | 1.44 | −2.00 |
| | 0.8 | 0.66 | 0.69 | 5.01 |
| | 0.4 | 0.34 | 0.31 | −10.01 |
| Turning | 12.5 | 11.23 | 11.61 | 0.75 |
| | 6.3 | 5.03 | 5.21 | 3.68 |
| | 3.2 | 2.58 | 2.52 | −2.39 |
| | 1.6 | 1.55 | 1.57 | 1.28 |
| | 0.8 | 0.77 | 0.79 | 3.02 |
| | 0.4 | 0.36 | 0.39 | 8.91 |
| Horizontal milling | 12.5 | 7.54 | 7.29 | −3.28 |
| | 6.3 | 5.32 | 5.28 | −0.69 |
| | 3.2 | 3.10 | 3.30 | 6.57 |
| | 1.6 | 1.90 | 1.81 | −4.66 |
| | 0.8 | 0.89 | 0.91 | 2.04 |
| Plane grinding | 1.6 | 1.48 | 1.42 | −3.95 |
| | 0.8 | 0.83 | 0.81 | −2.35 |
| | 0.4 | 0.42 | 0.41 | −2.08 |
| | **0.2** | 0.15 | 0.23 | **54.78** |
| | **0.1** | 0.12 | 0.17 | **38.59** |
| | **0.05** | 0.08 | 0.15 | **87.38** |
| External grinding | 1.6 | 1.52 | 1.20 | −21.16 |
| | 0.8 | 0.81 | 1.79 | −2.19 |
| | 0.4 | 0.54 | 0.48 | −11.11 |
| | **0.2** | 0.19 | 0.28 | **47.04** |
| Lapping | **0.1** | 0.1 | 0.18 | **79.67** |
| | **0.05** | 0.06 | 0.14 | **130.61** |

Note: The errors highlighted in red and the corresponding nominal values of the *Ra* parameter indicated on the roughness samples in Figure 14 are not shown.

Experimental data showed that the proposed system was able to measure surfaces with *Ra* values from 0.34 to 11.6 μm, which covers a common range of milling, turning, and grinding. In this range, measurement relative errors can be controlled within 10% (Figure 14).

The obtained result is consistent with that of Fu et al. [18], who showed that for very smooth surfaces with *Ra* values less than 0.4 μm, the laser confection system may have the following disadvantages, namely, a limited spot size and background noise. In addition, Fu et al. [18] found a measurement relative error of 5% for *Ra* 0.2 μm in the investigated range of *Ra* 0.2–6.35 μm, and the error of more than 50% obtained in our work in the investigated range of *Ra* 0.05–11.23 μm was most likely due to the different technical characteristics of the chromatic confocal sensor used. For the sensor LT-9010 M Keyence [18]: laser spot diameter = 2 μm and resolution = 0.01 μm, and for CL-PT010: laser spot diameter = 3.5 μm and resolution = 0.25 μm.

### 4.3. Correlation Analysis and Frequency Analysis for Development of the Estimated Functions—Power Spectrum Density and Autocorrelation Function

The predetermined law of the kinematic motion of the tool relative to the workpiece being processed assumes that as a result of processing, the ideal shape of the part with traces of the cutting tool will be obtained, which can be determined using kinematic and geometric relationships. For example, when turning a workpiece in its longitudinal section, there should be a series of arcs that correspond in shape to the profile of the cutter tip, and the distance between adjacent repeating arcs should be equal to the feed of the cutter for one turn of the workpiece. In this case, such predetermined irregularities would constitute theoretical roughness. However, the presence of dynamic processes on the

machine disturbs this predetermination and the real surface profile does not correspond to the ideal (theoretical) shape, built on the basis of geometric and kinematic calculations.

For various types of processing, the roughness of the machined surface is presented in the form of two components—deterministic, described by a periodic function, and random [38]—which together affect the topography of the surface. One of the reasons for the formation of a periodic, i.e., a systematic component of microroughness, is the trace of the cutting blade on the machined surface. As for the random component, it is problematic to theoretically predict its value, but it contains information about tool wear, random displacements of the tool relative to the part, as well as other random effects on the micro-profile of the machined surface during its shaping.

The purpose of this study was to develop a methodology for evaluating experimental data based on the concept of separating the total measurement information, for example, information characterizing the profile of the machined surface, into a systematic and random component of this information, using the mathematical apparatus of correlation analysis.

The mathematical model of the signal—profilograms—contains two components: systematic (deterministic signal) and random (stochastic signal). The same information of the profilogram will be presented in the time and frequency domains in accordance with the fast Fourier transform.

The mathematical models used of signals limited in time (or coordinates) were transformed to their "canonical" form, adopted in computer signal-processing technologies, by means of "centering" the corresponding time functions, i.e., by removing the constant or mean value and the expected value from the systematic and random components of the signals, respectively. In this case, the corresponding moments for centered random variables will be called the central moments of the distribution density of the random component of the signal. The first moment of the distribution density is the mathematical expectation of the signal. The second moment of the distribution density is the signal dispersion or the standard deviation of its amplitude from the mean value.

### 4.3.1. Correlation Analysis

The method for separating the signal characterizing the surface profilogram is as follows:

1.  The profilogram of the surface $y(x)$ is considered as a realization of a stationary random process on the length of the profilogram $l$, i.e., a random process with the constant expected value equal to zero.
2.  The real surface profile $Y(x)$ is represented as the sum of two components: deterministic (periodic, regular) and random (non-deterministic, aperiodic, irregular). In this case, the deterministic component $Y_\beta(x)$, in contrast to the random component, is a polyharmonic oscillation consisting of the sum of simple harmonics, and the random component $Y_\gamma(x)$ is a random stationary function with zero mathematical expectation and variance $D_\gamma = \sigma\gamma^2$. The mathematical model, for example of a profilogram signal, has the form [38]:

$$Y(x) = Y_\beta(x) + Y_\gamma(x) = \sum_{j=1}^{m} \left( A_j cos\omega_j x + B_j sin\omega_j x \right) + Y_\gamma(x)$$

where $A_j$, $B_j$ are the coefficients of the Fourier series, $j$ = 1, 2, ..., $n$ are serial numbers of harmonics with angular frequency $\omega_j$, and $Y_\beta(x)$ and $Y_\gamma(x)$ are deterministic and random centered variables.

Using the identification property of the correlation function of the total signal, the correlation analysis can be applied. In this case, the correlation function of the total signal $Y(x)$ is a tool for identifying the deterministic, $Y_\beta(x)$, and random, $Y_\gamma(x)$, components.

For the discrete implementation of the AutoCorrelation VI (LabVIEW Virtual Instrument), $Y$ represents a sequence whose indexing can be negative, $N$ is the number

of elements in the input sequence $X$, and the indexed elements of $X$ that lie outside its range are equal to zero according to the relationship: $x_j = 0$, $j < 0$ or $j \geq N$. Then, the AutoCorrelation VI obtains the elements of $Y$ using the following formula [38,39]:

$$Y_j = \sum_{k=0}^{N-1} X_k X_{j+k}$$

$$j = -(N-1), -(N-2), \ldots, -1, 0, 1, \ldots, (N-2), (N-1).$$

The elements of the output sequence $R_{xx}$ (the autocorrelation function of $X$) are related to the elements in the sequence $Y$ by [38,39]:

$$R_{xxi} = Y_{i-(N-1)}, \ i = 0, 1, 2, \ldots, 2(N-1).$$

In this case, the determination of the autocorrelation function was performed in the interval of double the length of the estimation.

Let us analyze the obtained correlograms of the signals characterizing the profilograms of the samples, representative of the periodic and aperiodic profiles (Figures 15–18):

-   The sample after turning (periodic profile) with a nominal roughness of $Ra$ = 1.6 (Figure 15) is sample No. 1.
-   The sample after plane grinding (aperiodic profile) with a nominal roughness of $Ra$ = 1.6 (Figure 17) is sample No. 2.

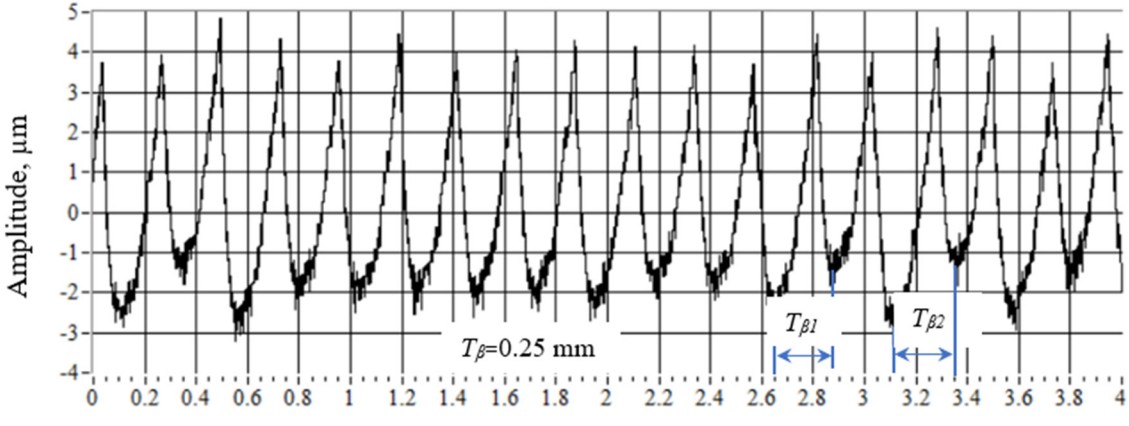

**Figure 15.** Surface profilogram of the sample No. 1 (turning $Ra$ = 1.6).

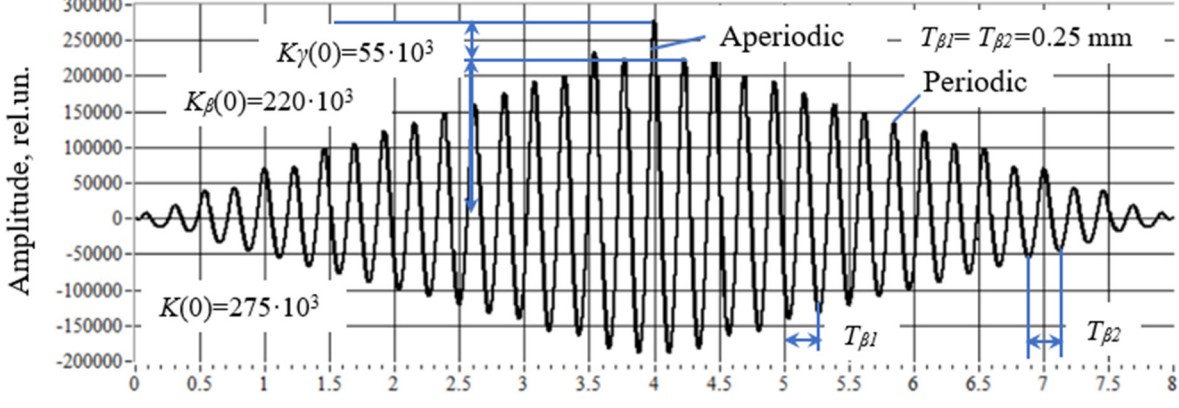

**Figure 16.** Correlogram of sample No. 1 (turning $Ra$ = 1.6).

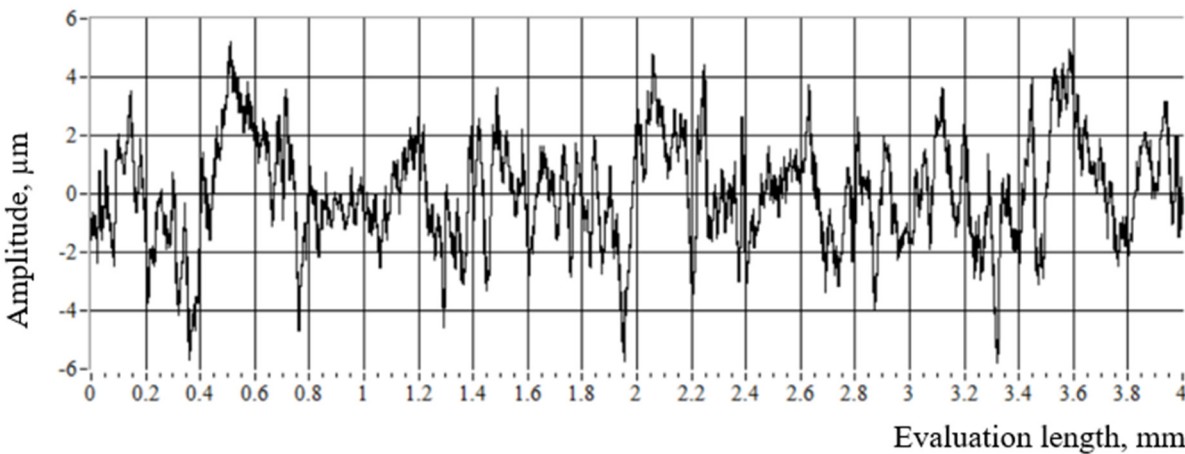

**Figure 17.** Surface profilogram of sample No. 2 (plane grinding *Ra* = 1.6).

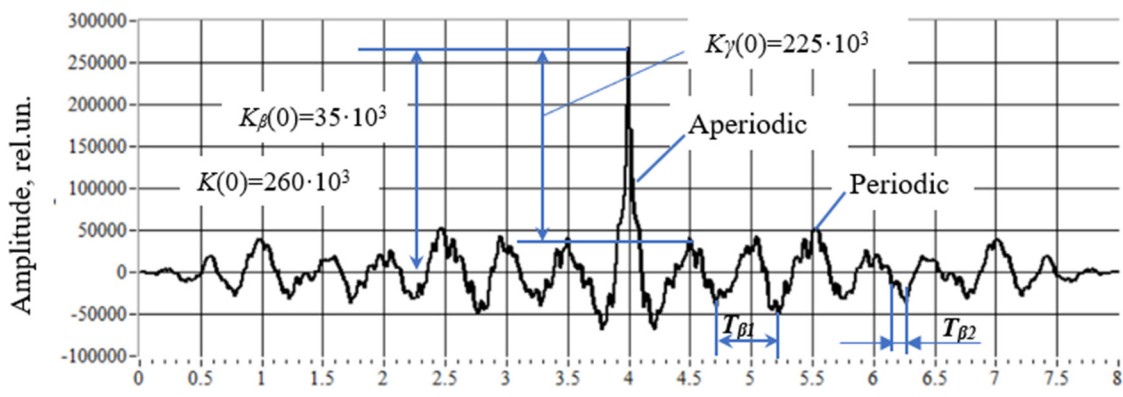

**Figure 18.** Correlogram of sample No. 2 (plane grinding *Ra* = 1.6).

1. Correlograms (Figures 16 and 18) were built on the double-estimation length ($l_n$ = 4 mm) and were symmetrical in the interval of 0–8 mm, relative to the reading $l_n$ = 4 mm. This corresponds to the well-known parity property of the autocorrelation function: $R_{xxi} = K(\tau)$.

2. With the abscissa $l_n$ = 4 mm, the autocorrelation function (Figures 16 and 18) takes its maximum value, equal to the sum of two variances: $K_\beta(0)$ of the deterministic component of the total signal and the variance of $K_\gamma(0)$ of the random component of the signal, i.e., $K(0) = K_\beta(0) + K_\gamma(0)$.

   For samples No. 1 and No. 2, these equations have the form:
   $275 \times 10^3 = 220 \times 10^3 + 55 \times 10^3$ and $260 \times 10^3 = 225 \times 10^3 + 35 \times 10^3$, respectively.

   That is, the deterministic component of the signal in the first case (sample No. 1, Figure 15) was 4 times larger than its random component and was 80% of $K(0)$. In the second case (sample No. 2, Figure 17), on the contrary, the random component of the profilogram signal exceeded its deterministic one by 6.4 times. Moreover, it was 85% of $K(0)$.

3. The wave step (0.25 mm) of the initial total signal for sample No. 1 (Figure 15) coincided with the wave step of the autocorrelation function for the same sample, i.e., $T_{\beta 1} = T_{\beta 2} = 0.25$ mm (Figure 16). At the same time, for sample No. 2 (Figure 17), the deterministic component of the signal associated with its periodicity was not expressed—it was hidden and characterized by variable steps $T_{\beta 1} = 0.5$ mm and $T_{\beta 2} = 0.1$ mm (Figure 18).

For other samples obtained by turning and plane grinding, the relationship between the deterministic and random components is presented in Table 4.

**Table 4.** Ratio of deterministic and random components.

| | Machining Operation | | | | |
|---|---|---|---|---|---|
| | Turning | | | Plane Grinding | |
| *Ra*, μm | Deterministic Component, % | Random Component, % | *Ra*, μm | Deterministic Component, % | Random Component, % |
| 0.4 | 28.6 | 71.4 | 0.4 | 15 | 85 |
| 0.8 | 55 | 45 | 0.8 | 22 | 78 |
| 1.6 | 80 | 20 | 1.6 | 15 | 85 |
| 3.2 | 85 | 15 | | | |
| 6.3 | 89.7 | 10.3 | | | |
| 12.5 | 89.3 | 10.7 | | | |

Table 4 shows the following:

1.  With an increase of the nominal roughness from *Ra* 0.4 to *Ra* 12.5 (in this case, the actual values changed from 0.363 μm to 11.23 μm, measured by SJ-400) for samples after the turning operation, the proportion of the deterministic component in the total signal increased.
2.  Of the cutting modes, the feed and cutting speed had the most significant influence. Turning formed a periodic profile on the sample surface by changing the tool feed.

The height and step of the irregularities were formed by the feed of the tool. It is this that forms the deterministic component. With an increase in the feed, the surface roughness, the height of the irregularities, and the step between the irregularities, equal to the feed rate per one revolution of the part, increased. Therefore, with an increase in the nominal roughness parameter *Ra* from 0.4 microns to 12.5 microns (Table 4), the proportion of the deterministic component increased.

3.  With an increase of the nominal roughness from *Ra* 0.4 to *Ra* 1.6 (in this case, the actual values changed from 0.42 μm to 1.48 μm, measured SJ-400) for samples after the grinding operation, the ratio between the deterministic and random components was almost constant.
4.  Grinding is a stochastic process, during which the surface roughness was formed by randomly arranged abrasive grains. Consequently, an aperiodic profile was formed on the surface of the sample. Therefore, the random component was larger than the deterministic one, and the ratio between them practically remained constant.

### 4.3.2. Frequency Analysis

In the study of oscillations, the frequency approach, which represents the possibility of transferring information from a temporal form of its representation to a frequency one, has been used for a long time. This is because the vibroacoustic vibration parameters (vibration acceleration, vibration velocity, vibration displacement), by their physical nature, change in time as the workpiece is processed, in contrast to the signal of surface irregularities, which is a function of one (one-dimensional profilogram) or two (topography) coordinates of the machined surface.

The purposes of the study were:

-   To develop a frequency profile analysis technique based on the fast Fourier transform (FFT) and the formation of power spectrum density.
-   To consider the possibility of using the power spectrum density spectrogram to analyze the situation during turning. This setting can be effective as it provides a significant signal gain over the noise introduced into the system. This approach can be used to develop software for process-monitoring systems.

To do this, in the NI-LabVIEW program, the output signal of the profilograms (after the chromatic confocal sensor measurement) for turning samples with *Ra* 0.4 (Figure 19a), *Ra* 1.6 (Figure 20a), and *Ra* 12.5 (Figure 21a) was sent to the Spectral Measurements block of NI-LabVIEW, which performed the FFT procedure. The results obtained after the FFT

in the form of a spectrogram were displayed on the front panel of the NI-DAQmx virtual instrument using the Waveform Graph display blocks (Figures 19b, 20b and 21b).

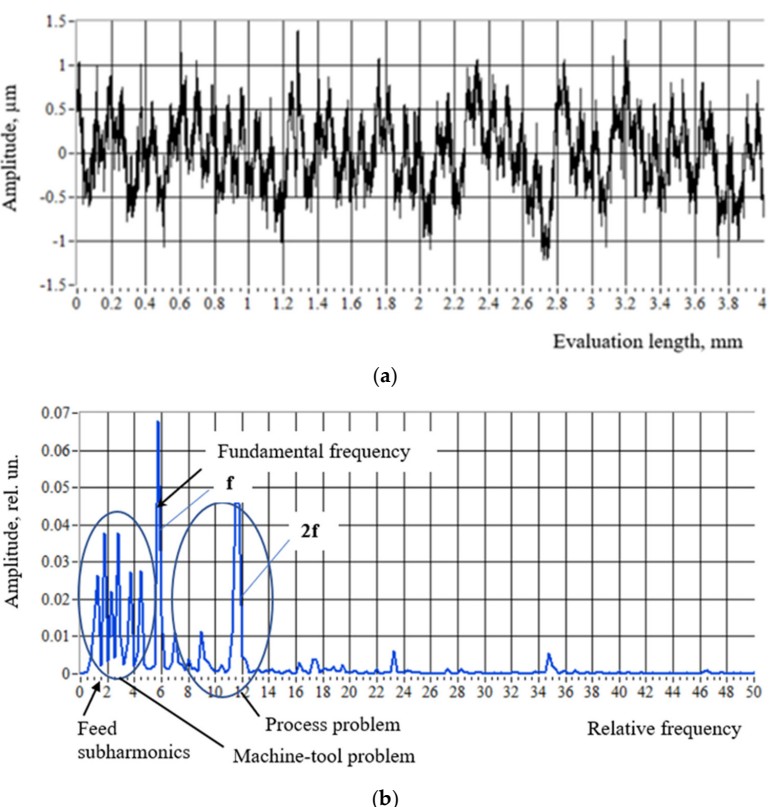

**Figure 19.** (**a**) Surface profilogram of the sample—turning *Ra* 0.4, and (**b**) its power spectrum density.

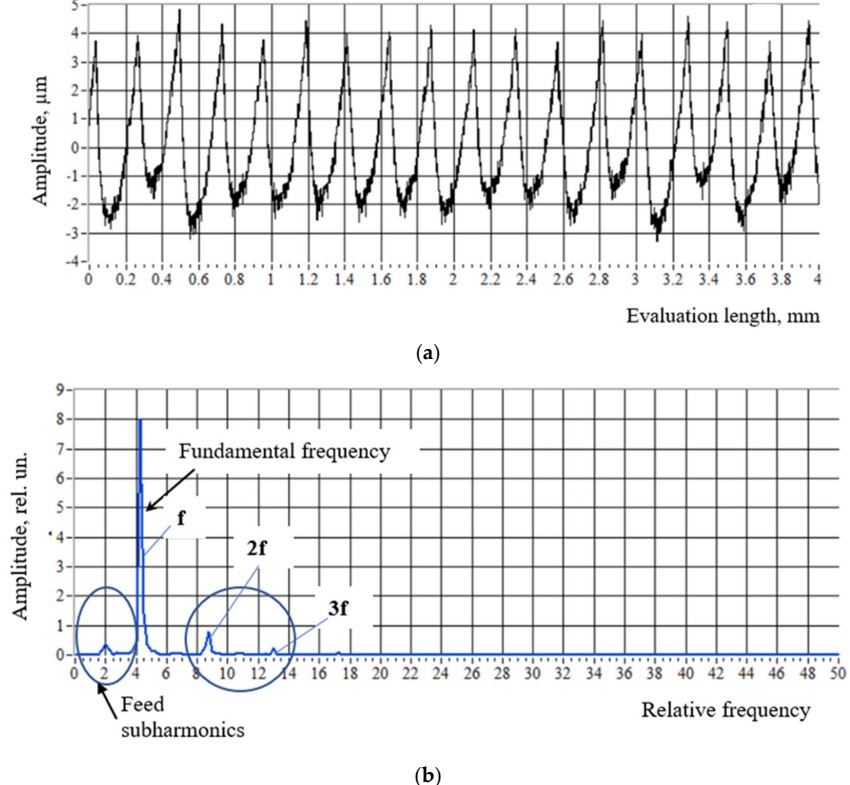

**Figure 20.** (**a**) Surface profilogram of the sample—turning *Ra* 1.6, and (**b**) its power spectrum density.

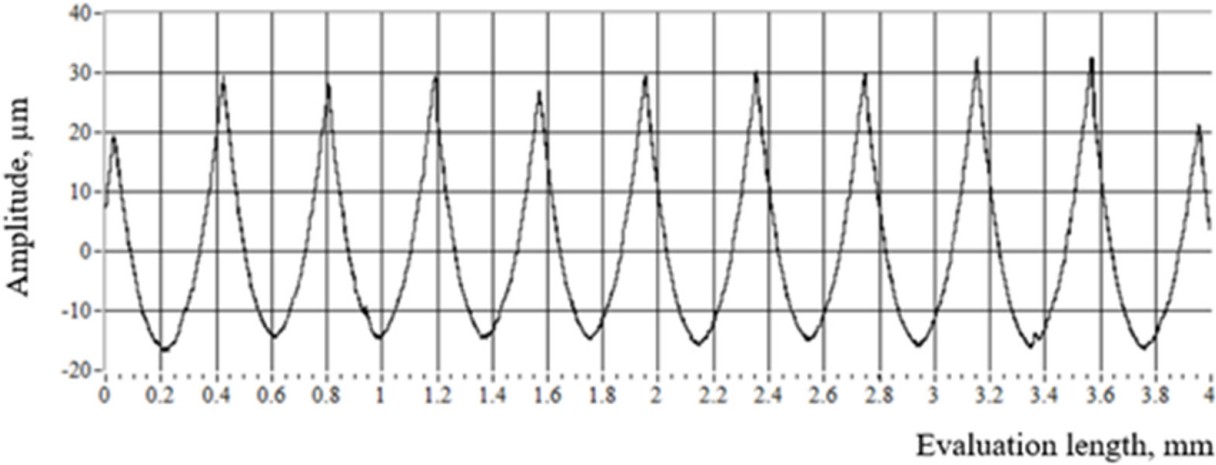

(**a**)

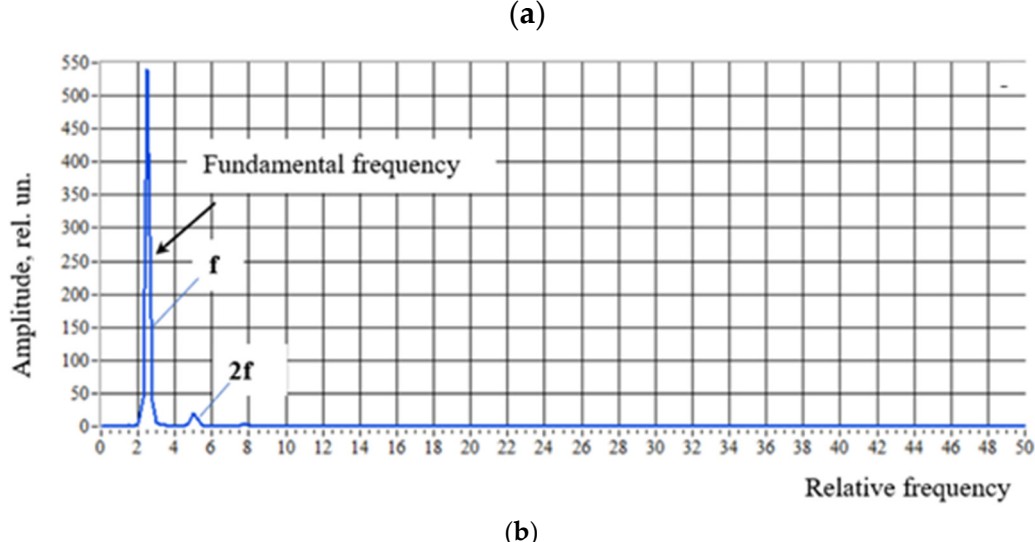

(**b**)

**Figure 21.** (**a**) Surface profilogram of the sample—turning *Ra* 12.5, and (**b**) its power spectrum density.

Figures 19b, 20b and 21b show that the PSD spectra differed in the set of harmonic components and the fundamental harmonic amplitude. For example, the PSD spectrum in Figure 19b is characterized by the appearance to the left of the main wave of oscillations, the wavelengths of which were greater than the fundamental wavelength. This may be due to the wear of bearings and the insufficient stiffness of the machine, which causes vibration [4].

In accordance with Whitehouse's hypothesis [4], the spectrum can be divided into two parts: the first part is to the left of the main part, and the second part is to the right of the main part. A process problem appears on the right side, and a machine problem appears on the left. There are interesting assumptions about the use of PSD for diagnosing cutter wear during turning and the autocorrelation function for diagnosing wheel wear during grinding. It is necessary to compare the PSD spectra obtained by a good tool. The spectrum should reflect the fundamental frequency corresponding to the feed of the tool and several harmonics due to the shape of the tool. As the tool became worn, the ratio of harmonic amplitudes to the fundamental harmonic amplitude increased. In addition, the formation of chips and surface cracks can be reflected in the spectrum [4]. However, this issue will be discussed in future research.

## 5. Conclusions

1.  A measuring system based on a chromatic confocal sensor was developed that allows measuring surface roughness with the ability to select settings, such as: the speed of the measuring head, the evaluation length, the traverse length, the cutoff, the number of tracks on the sample surface, and the distance between the measured tracks.

2.  When operation testing of the developed measuring system was conducted on a standard metal sample with a known roughness, it was found that with an increase of the speed of the measuring head, the error in the estimates of *Ra* and *Rz* increased. An error level of 5% corresponded to the probe movement speed of $V = 0.2$ mm/s.

3.  The roughness parameters of metal samples are invariant to the speed of the tip of the stylus profilometer, Portable Surface Roughness Tester Surftest SJ-400. As the handpiece speed increased from 0.05 mm/s to 1 mm/s, *Ra* changed by 0.43% and *Rz* by 0.2%. Therefore, the maximum speed of the probe can be set during the measurement.

4.  When testing the operation of the developed measuring system using the surface roughness comparator standards composite set, the assembled measuring system based on the chromatic confocal sensor showed its performance in assessing the roughness parameter *Ra* from 0.4 μm to more than 12 μm, which covers a common range of milling, turning, and grinding. In this range of measurement, relative errors can be controlled within 10%. Moreover, in the *Ra* range from 2 μm to 12 μm, the error was 5%.

5.  Under these laboratory conditions, in the studied range of scanning speeds for a given range of changes in the roughness parameters of samples obtained by mechanical processing methods, the measurement uncertainty was not detected.

6.  Frequency analysis and correlation analysis of profilograms were carried out. Within the framework of correlation analysis, a method for separating the deterministic and random components of the processed signal of the surface profile was developed and tested on examples. The method allowed identifying the hidden periodicity of the profile. Relationships between the deterministic and random components were established based on the example of samples processed by turning and grinding.

Frequency analysis, for example, carried out on samples obtained by turning, made it possible to identify different states of the turning technological system. For example, the PSD spectrum contains the fundamental frequency and may also contain harmonics that indicate problems with the machine and problems with the process. However, this issue requires further research.

Based on this study, the focus of further research will be devoted to the development of diagnostic features for process monitoring based on profilogram estimates such as the autocorrelation function and the power spectrum density. It is also necessary to carry out studies on the measurement uncertainty of the roughness parameters during in situ measurements. In this case, it becomes necessary to take into account the measurement noise added to the output signal. It is important to identify the sources of uncertainty (e.g., vibrations, temperature fluctuations, scanning speed, surface contamination, sample reflectivity) in a particular production and their contribution to the measurement uncertainty.

**Author Contributions:** Conceptualization, N.L.; methodology, N.L.; software, M.C.; validation, N.L.; formal analysis, G.E.O.; investigation, N.L.; writing—original draft preparation, N.L.; writing—N.L. and G.E.O.; visualization, N.L. and M.C.; supervision, G.E.O.; project administration, G.E.O. All authors have read and agreed to the published version of the manuscript.

**Funding:** This research work was funded by Enterprise Ireland and the European Commission under the Horizon 2020 Marie Skłodowska-Curie Actions.

**Institutional Review Board Statement:** Not applicable.

**Informed Consent Statement:** Not applicable.

**Data Availability Statement:** Not applicable.

**Acknowledgments:** This research was carried out at the Department of Mechanical, Manufacturing, and Biomedical Engineering of Trinity College of Dublin in accordance with the project "Data driven optimization of complex-shaped parts produced by grinding and abrasive processes", funded by Enterprise Ireland and the European Commission under the Horizon 2020 Marie Skłodowska-Curie Actions.

**Conflicts of Interest:** The authors declare no conflict of interest. The funders had no role in the design of the study; in the collection, analyses, or interpretation of data; in the writing of the manuscript, or in the decision to publish the results.

## Appendix A

**Table A1.** *Ra* and *Rz* values of the precision roughness reference specimen using SJ-400.

| Number of Tracks | \multicolumn{6}{}{Number of Tests} | | | | | | Average | | Sample Standard Deviation, µm | | Margin of Error, µm | |
|---|---|---|---|---|---|---|---|---|---|---|---|---|
| | 1 | | 2 | | 3 | | | | | | | |
| | *Ra*, µm | *Rz*, µm | *Ra*, µm | *Rz*, µm | *Ra*, µm | *Rz*, µm | *Ra*, µm | *Rz*, µm | *Ra* | *Rz* | *Ra* | *Rz* |
| \multicolumn{13}{}{Measuring Speed, mm/s — 0.05} | | | | | | | | | | | |
| 1 | 2.98 | 9.50 | 2.98 | 9.50 | 2.98 | 9.50 | 2.98 | 9.50 | 0 | 0.115 | 0 | 0.287 |
| 2 | 2.98 | 9.60 | 2.98 | 9.60 | 2.98 | 9.60 | 2.98 | 9.60 | 0 | 0 | 0 | 0 |
| 3 | 2.98 | 9.50 | 2.99 | 10.10 | 2.99 | 9.70 | 2.99 | 9.77 | 0.006 | 0.306 | 0.014 | 0.759 |
| | | | | | | | **2.983** | **9.62** | **0.006** | **0.137** | **0.014** | **0.339** |
| \multicolumn{13}{}{0.1} | | | | | | | | | | | |
| 1 | 2.98 | 9.50 | 2.98 | 9.50 | 2.98 | 9.50 | 2.98 | 9.50 | 0 | 0 | 0 | 0 |
| 2 | 2.98 | 9.70 | 2.98 | 9.60 | 2.98 | 9.60 | 2.98 | 9.63 | 0 | 0.058 | 0 | 0.143 |
| 3 | 2.99 | 9.50 | 3.00 | 10.10 | 2.99 | 9.50 | 2.99 | 9.70 | 0.006 | 0.346 | 0.014 | 0.861 |
| | | | | | | | **2.983** | **9.61** | **0.006** | **0.101** | **0.014** | **0.252** |
| \multicolumn{13}{}{0.5} | | | | | | | | | | | |
| 1 | 2.98 | 9.50 | 2.97 | 9.50 | 2.97 | 9.50 | 2.97 | 9.50 | 0 | 0 | 0 | 0 |
| 2 | 2.98 | 9.70 | 2.98 | 9.60 | 2.98 | 10.1 | 2.98 | 9.80 | 0 | 0.265 | 0 | 0.657 |
| 3 | 2.98 | 9.50 | 2.98 | 9.50 | 2.98 | 9.50 | 2.98 | 9.50 | 0 | 0 | 0 | 0 |
| | | | | | | | **2.977** | **9.60** | **0.006** | **0.173** | **0.014** | **0.430** |
| \multicolumn{13}{}{1.0} | | | | | | | | | | | |
| 1 | 2.97 | 9.50 | 2.99 | 10.30 | 2.97 | 9.50 | 2.98 | 9.77 | **0.012** | **0.462** | 0.029 | **1.147** |
| 2 | 2.97 | 10.50 | 2.97 | 9.70 | 2.97 | 9.70 | 2.97 | 9.97 | 0 | **0.462** | 0 | **1.147** |
| 3 | 2.98 | 9.50 | 2.98 | 9.50 | 2.98 | 9.50 | 2.98 | 9.50 | 0 | 0 | 0 | 0 |
| | | | | | | | **2.97** | **9.60** | **0.006** | **0.236** | **0.014** | **0.586** |

Formulas for the margin of error for comparing the results (SR-Test, SJ-400) and the impact speed (SR-Test):
Sample standard deviation is:

$$s = \sqrt{\frac{\sum\left(x - \bar{x}\right)^2}{n-1}}$$

where $x$ is the parameter current value, $\bar{x}$ is the sample mean, and $n$ is the sample size.
The margin of error is:

$$E = t_{\alpha/2} \cdot \frac{s}{\sqrt{n}}$$

$t_{\alpha/2}$—t-score.
In our case, $n = 3$, and $t_{\alpha/2} = 4.303$, at a confidence level of 95%, which means that:

$$0.95 = 1 - \alpha$$

The significance level $\alpha = 1 - 0.95 = 0.05$, and the degrees of freedom $n - 1 = 3 - 1 = 2$.

**Table A2.** *Ra* and *Rz* values of the precision roughness reference specimen using the chromatic confocal sensor.

| Number of Tracks | \multicolumn{6}{}{Number of Tests} | | | | | | Average | | Sample Standard Deviation, µm | | Margin of Error, µm | |
|---|---|---|---|---|---|---|---|---|---|---|---|---|
| | 1 | | 2 | | 3 | | | | | | | |
| | *Ra*, µm | *Rz*, µm | *Ra*, µm | *Rz*, µm | *Ra*, µm | *Rz*, µm | *Ra*, µm | *Rz*, µm | *Ra* | *Rz* | *Ra* | *Rz* |
| \multicolumn{13}{}{Measuring Speed, mm/s — 0.05} | | | | | | | | | | | |
| 1 | 3.022 | 10.524 | 3.030 | 10.613 | 3.006 | 10.525 | 3.019 | 10.554 | 0.012 | 0.051 | 0.030 | 0.126 |
| 2 | 2.768 | 10.621 | 2.769 | 10.602 | 2.771 | 10.583 | 2.769 | 10.603 | 0.002 | 0.019 | 0.003 | 0.046 |
| 3 | 2.931 | 10.411 | 2.930 | 10.479 | 2.934 | 10.597 | 2.932 | 10.496 | 0.002 | 0.939 | 0.006 | 0.233 |
| | | | | | | | **2.907** | **10.550** | **0.127** | **0.053** | **0.315** | **0.133** |
| \multicolumn{13}{}{0.1} | | | | | | | | | | | |
| 1 | 2.930 | 10.373 | 2.969 | 10.331 | 2.973 | 10.260 | 2.957 | 10.321 | 0.024 | 0.057 | 0.059 | 0.142 |
| 2 | 2.767 | 10.409 | 2.755 | 10.457 | 2.788 | 10.288 | 2.769 | 10.385 | 0.017 | 0.087 | 0.042 | 0.216 |
| 3 | 2.915 | 10.422 | 2.912 | 10.414 | 2.896 | 10.398 | 2.908 | 10.411 | 0.011 | 0.013 | 0.026 | 0.031 |
| | | | | | | | **2.878** | **10.372** | **0.097** | **0.046** | **0.240** | **0.115** |
| \multicolumn{13}{}{0.15} | | | | | | | | | | | |
| 1 | 2.897 | 10.263 | 2.958 | 10.314 | 2.951 | 10.487 | 2.935 | 10.354 | 0.033 | 0.117 | 0.083 | 0.291 |
| 2 | 2.737 | 10.283 | 2.739 | 10.342 | 2.752 | 10.282 | 2.743 | 10.302 | 0.008 | 0.034 | 0.019 | 0.084 |
| 3 | 2.848 | 10.303 | 2.846 | 10.261 | 2.837 | 10.341 | 2.844 | 10.302 | 0.006 | 0.04 | 0.014 | 0.099 |
| | | | | | | | **2.840** | **10.319** | **0.116** | **0.030** | **0.288** | **0.076** |
| \multicolumn{13}{}{0.20} | | | | | | | | | | | |

**Table A2.** *Cont.*

| Number of Tracks | Number of Tests 1 | | 2 | | 3 | | Average | | Sample Standard Deviation, μm | | Margin of Error, μm | |
|---|---|---|---|---|---|---|---|---|---|---|---|---|
| | *Ra*, μm | *Rz*, μm | *Ra*, μm | *Rz*, μm | *Ra*, μm | *Rz*, μm | *Ra*, μm | *Rz*, μm | *Ra* | *Rz* | *Ra* | *Rz* |
| **Measuring Speed, mm/s 0.05** | | | | | | | | | | | | |
| 1 | 2.891 | 10.326 | 2.891 | 10.236 | 2.870 | 10.145 | 2.884 | 10.235 | 0.012 | 0.091 | 0.030 | 0.226 |
| 2 | 2.707 | 10.168 | 2.745 | 9.954 | 2.810 | 9.729 | 2.754 | 9.950 | 0.052 | 0.219 | 0.129 | 0.546 |
| 3 | 2.811 | 10.160 | 2.832 | 10.106 | 2.827 | 10.138 | 2.823 | 10.135 | 0.011 | 0.028 | 0.028 | 0.068 |
| | | | | | | | **2.821** | **10.107** | **0.078** | **0.144** | **0.194** | **0.359** |
| **0.25** | | | | | | | | | | | | |
| 1 | 2.815 | 10.156 | 2.861 | 10.189 | 2.803 | 10.079 | 2.826 | 10.141 | 0.031 | 0.056 | 0.076 | 0.139 |
| 2 | 2.734 | 9.839 | 2.708 | 9.953 | 2.742 | 9.560 | 2.728 | 9.784 | 0.018 | 0.202 | 0.044 | 0.502 |
| 3 | 2.772 | 9.989 | 2.764 | 10.047 | 2.760 | 10.034 | 2.765 | 10.023 | 0.006 | 0.030 | 0.014 | 0.076 |
| | | | | | | | **2.773** | **9.983** | **0.065** | **0.182** | **0.161** | **0.452** |
| **0.35** | | | | | | | | | | | | |
| 1 | 2.665 | 9.794 | 2.740 | 9.955 | 2.692 | 9.751 | 2.699 | 9.833 | 0.094 | 0.268 | 0.038 | 0.108 |
| 2 | 2.690 | 9.196 | 2.714 | 9.588 | 2.603 | 9.335 | 2.669 | 9.373 | 0.145 | 0.493 | 0.058 | 0.199 |
| 3 | 2.649 | 9.651 | 2.716 | 9.487 | 2.648 | 9.530 | 2.671 | 9.556 | 0.098 | 0.210 | 0.039 | 0.085 |
| | | | | | | | **2.679** | **9.587** | **0.024** | **0.232** | **0.059** | **0.576** |
| **0.45** | | | | | | | | | | | | |
| 1 | 2.544 | 9.219 | 2.593 | 9.538 | 2.603 | 8.989 | 2.579 | 9.249 | 0.032 | 0.275 | 0.078 | 0.684 |
| 2 | 2.490 | 8.749 | 2.579 | 8.885 | 2.452 | 8.815 | 2.507 | 8.816 | 0.066 | 0.068 | 0.163 | 0.168 |
| 3 | 2.599 | 8.738 | 2.545 | 8.764 | 2.517 | 9.089 | 2.554 | 8.863 | 0.042 | 0.196 | 0.103 | 0.487 |
| | | | | | | | **2.547** | **8.976** | **0.040** | **0.237** | **0.100** | **0.589** |
| **0.55** | | | | | | | | | | | | |
| 1 | 2.372 | 8.718 | 2.456 | 8.682 | 2.352 | 8.656 | 2.393 | 8.686 | 0.055 | 0.032 | 0.136 | 0.078 |
| 2 | 2.300 | 8.188 | 2.345 | 8.163 | 2.364 | 8.497 | 2.336 | 8.282 | 0.033 | 0.186 | 0.081 | 0.462 |
| 3 | 2.348 | 8.549 | 2.358 | 8.338 | 2.330 | 8.476 | 2.346 | 8.454 | 0.0142 | 0.107 | 0.035 | 0.266 |
| | | | | | | | **2.358** | **8.474** | **0.043** | **0.202** | **0.106** | **0.502** |
| **0.65** | | | | | | | | | | | | |
| 1 | 2.149 | 7.974 | 2.226 | 7.405 | 2.291 | 7.686 | 2.222 | 7.688 | 0.071 | 0.285 | 0.176 | 0.707 |
| 2 | 2.249 | 7.668 | 2.122 | 7.469 | 2.129 | 7.428 | 2.167 | 7.522 | 0.071 | 0.128 | 0.178 | 0.319 |
| 3 | 2.177 | 7.454 | 2.149 | 7.651 | 2.216 | 7.668 | 2.181 | 7.591 | 0.033 | 0.119 | 0.083 | 0.297 |
| | | | | | | | **2.189** | **7.600** | **0.039** | **0.083** | **0.098** | **0.208** |
| **0.75** | | | | | | | | | | | | |
| 1 | 1.938 | 7.074 | 1.918 | 7.306 | 1.933 | 7.179 | 1.929 | 7.187 | 0.011 | 0.116 | 0.027 | 0.288 |
| 2 | 1.919 | 7.179 | 1.941 | 7.013 | 2.055 | 7.086 | 1.972 | 7.093 | 0.073 | 0.083 | 0.181 | 0.207 |
| 3 | 1.993 | 7.091 | 1.977 | 6.665 | 1.991 | 7.000 | 1.987 | 6.919 | 0.008 | 0.224 | 0.021 | 0.557 |
| | | | | | | | **1.963** | **7.066** | **0.041** | **0.136** | **0.101** | **0.337** |
| **1** | | | | | | | | | | | | |
| 1 | 1.412 | 4.961 | 1.325 | 5.144 | 1.337 | 5.064 | 1.358 | 5.056 | 0.047 | 0.092 | 0.117 | 0.228 |
| 2 | 1.317 | 4.743 | 1.305 | 5.056 | 1.313 | 5.017 | 1.312 | 4.939 | 0.006 | 0.170 | 0.015 | 0.423 |
| 3 | 1.354 | 5.358 | 1.348 | 5.281 | 1.336 | 4.711 | 1.346 | 5.117 | 0.009 | 0.354 | 0.023 | 0.878 |
| | | | | | | | **1.339** | **5.037** | **0.024** | **0.090** | **0.058** | **0.224** |

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
