# Peer review of "Contactless Method for Measurement of Surface Roughness Based on a Chromatic Confocal Sensor"

_machines, doi:10.3390/machines11080836_

Round 1

Reviewer 1 Report

In this work, the author has proposed a non-contact in-situ surface roughness measuring system. The proposed system utilizes a chromatic confocal sensor. The assembled measuring system based on the chromatic confocal laser sensor shows its performance in assessing the roughness parameters Ra from more than 0.34 μm to 12 μm which covers a common range of milling, turning and grinding. In this range measurement relative errors can be controlled within 10%. Furthermore, frequency analysis and correlation analysis of profilograms are performed. The results of the analysis can be further used to develop diagnostic functions for process monitoring based on profilogram estimates, such as the autocorrelation function and power spectrum density. I would like to give some comments and suggestions. The detailed comments are as follows:

1) The authors write: “Only one selected frequency from the beam incident on the surface is focused on the passive optical system, which depends on the height of each point and gives a clear image of this point by one photodetector. The photodetector is an accurate spectrometer that allows you to determine the wavelength that gives information about the height of the measurement roughness [3].” What is the light source used in the optical system? Can the LEDs be used as the white light source in confocal sensor? The authors should mention the light source used in optical system. To give the readers a much broader view, recent literature related to white light source based on LEDs, such as Laser & Photonics Reviews 2023, 17, 2200455 (https://doi.org/10.1002/lpor.202200455); Optics Express 27(12), A669 (2019), etc. should be added, so that the readers can be clear about the state-of-the-art of this topic.  

2) In page 10, conclusion 3 from Table A1 shows that sample standard deviation and margin of error parameters for the Ra parameter remain constant when the speed of the measuring head increases. Conclusion 1 from Table A2 shows that when the speed of the measuring head increases, the sample standard deviation and margin of error parameters for Ra and Rz decrease. What is the reason for the different phenomenon?

3) Line 395, the author says that in this range measurement relative errors can be controlled within 10% (Fig.14). However, from table 3 we can know that some errors can exceed 10%, even can reach 130%.

4) Line 467, what does Yj, Xk, and Xj+k stand for? The author should explain them. 

5) In table 1, the values of resolution and positioning are 0.000125 μm. I think the unit is wrong. Please check it.

6) The author mentions that a non-contact in-situ surface roughness measuring system is proposed in this paper. Can the author provide evidences about this since the “in-situ” property has not been found in the paper?

7) Line 494, the deterministic component of the signal associated with its periodicity is hidden. Why did the author choose the variable steps ??1 and ??2 as 0.5 mm and 0.1‍‍ mm, respectively?

8) Why did not the author do the frequency analysis of sample No. 2 (plane grinding)? 

9) Some mistakes are found. For example, in line 326, there is an extra “-” between the words parameter and from. In figure 18, “Ky(0) = 225·10” should be “Ky(0) = 225·103”. In table A2, the symbol of the decimal point is ‘.’ rather than ‘,’. Please proofread your manuscript carefully.

Author Response

Best regards,

Natalia Lishchenko.

Reviewer 2 Report

This paper studied on the "Contactless Measurement Surface Roughness based on Chromatic Confocal Sensor"

(1) Recently, there are two types of surface roughness measurement method (i.e., contact type and non contact type). These methods are possible for measuring the roughness value of materials. Why are "Contactless Measurement" type needed? Is it more better than the previous types?

(2) In introduction section, the author should added more researches that are related to the surface roughness "measurement method". The main highlight and purpose should be clearly written in this section.

(3) In Figure 2 the "principle work of the confocal sensor", the quality of image is too low. Please use the 3 dimensional view of "principle work of the confocal sensor" for the Figure 2.

(4) The letters in Figure 11 are too small. Increase font size in Figure 11. 

- Moderate editing of English language required.

Author Response

Sincerely,

Natalia Lishchenko.

Reviewer 3 Report

Dear Author(s), the manuscript ‘Contactless Measurement Surface Roughness based on Chro-2 matic Confocal Sensor’, Manuscript ID: machines-2495506, have some weakness that must be revised appropriately.

Please find below some, of the most crucial comments:

1.      About the title, I would consider some modifications that the Author(s) proposed regarding some methods based on the Confocal measurement. From that matter, it should be emphasized in the title that the new method is applied and validated by adding the word method, approach or other, etc.

2.      In the second section, Literature Review, there is no clear critical review, especially indicating the weaknesses of the non-contact methods. The advantages of confocal measurement over some optical methods should be presented more precisely. Currently, the Reader is not conveyed what the main overcome is. Respectively, more disadvantages of other measuring techniques must be designated.

3.      Still in the second section. The number of cited references is relatively small. If the Author(s) propose the ‘Literature review’ in the subject area, there must be more items included, e.g.:

Contact and non-contact measurement comparison:

(1)   https://www.doi.org/10.1016/0043-1648(95)06697-7

(2)   https://www.doi.org/10.1016/j.ijleo.2023.170919

Autocorrelation Function and Power Spectral Density analysis:

(3)   https://www.doi.org/10.3390/coatings13010074

(4)   https://www.doi.org/10.1063/1.1914947

(5)   https://www.doi.org/10.1016/j.triboint.2017.02.016

4.      The description of the stylus and non-contact instruments are not derived with any information on the uncertainty or noise. The main purpose of the paper is to propose a new method for fast manufacturing measurement, based on saving measurement accuracy, however, the precision is not presented comprehensively. Please look for some additional information on the uncertainty and noise analysis:

(6)   https://www.doi.org/10.1088/2051-672X/3/3/035004

(7)   https://www.doi.org/10.3390/ma15155137

(8)   https://www.doi.org/10.1088/0957-0233/23/3/035008

5.      The main idea of the stylus and contactless measurement is not clear, presented in subsection 4.1. It is rather obvious that contact and non-contact measurement methods gave many different results. In the case of comparison, one type of measurement would always give some advantages over the other. From that matter, the motivation, even previously widely described, is not clear.

6.      The correlation analysis, presented in subsection 4.3.1., should be separated from already well-known studies. For example, the equations in lines 458 and 467 and many sub-equation in the text, should be referenced to the primary sources if are not first proposed. Generally, in this subsection, it is not clear what is the novelty and what was already presented (published previously).

7.      What is the main advantage of the application of FFT and PSD studies? In subsection 4.3.2., the motivation is unknown or, at least, not clear. Looks like refereeing to the already known (for more than 20 years) general usefulness of the mentioned functions.

8.      The ‘Conclusion’ section is too long and the Reader feels confused. Secondly, the main purpose is hidden from some details. Finally, it is difficult what exactly the Author(s) are trying to convey. Please make this section more valuable and concrete.

Moreover, some additional, editorial issues must be raised as well:

9.      Additionally, the third section has the number ‘2’ instead of ‘3’.

10.  The formatting of the ‘References’ must be unified according to the journal template requirements. Moreover, I’m not sure if all of the existing DOI links were provided.

Generally, the proposed manuscript can be classified as interesting, however, has some weaknesses, respectively, at least in the current form, and is not suitable for publication in a quality journal as the Machines is.

The manuscript must be improved significantly before any further processing, if allowed by the Editor.

Author Response

Sincerely,

Natalia Lishchenko.

Round 2

Reviewer 2 Report

The quality of paper has been improved. It can be accepted.

Moderate editing of English language required

Reviewer 3 Report

Dear Authors, the manuscript was improved according to the raised comments so can be further processed by the Editorial Office of the Machines journal.